

# Impact of shrub branches on the shortwave vertical irradiance profile in snow

Florent Domine[1,2,3*], Mireille Quémener[4,5,6*], Ludovick Bégin[4,5,6], Benjamin Bouchard[2,7,8], Valérie Dionne[3,4], Sébastien Jerczynski[4,5,6], Raphaël Larouche[1,5], Félix Lévesque-Desrosiers[1,2,5], Simon-Olivier Philibert[3,4], Marc-André Vigneault[4,5,6], Ghislain Picard[9], Daniel C. Côté[4,5,6]

[1] Takuvik Joint International Laboratory, Université Laval (Canada) and CNRS-INSU (France), Québec City, Canada.
[2] Centre d'Études Nordiques, Université Laval, Québec City, Canada.
[3] Department of Chemistry, Université Laval, Québec City, Canada.
[4] CERVO Brain Research Center, Québec (QC), Canada
[5] Département de physique, de génie physique et d'optique, Université Laval, Québec (QC), Canada
[6] Center for Optics, Photonics and Lasers, Québec (Qc), Canada
[7] Department of Civil and Water Engineering, Université Laval, Quebec City, Canada
[8] CentrEau – Water Research Center, Université Laval, Quebec City, Canada
[9] Université Grenoble Alpes, CNRS, IRD, Grenoble INP, IGE, Grenoble, France

Correspondence to:      Florent Domine( florent.domine@gmail.com)
                        Mireille Quémener (mireille.quemener@cervo.ulaval.ca)

**Abstract**. In the Arctic, shrubs are expanding and are covered by snow most of the year. Shrub branches buried in snow absorb solar radiation and therefore reduce irradiance. This reduces photochemical reaction rates and the emission of reactive and climatically active molecules to the atmosphere. Here we monitored irradiance at selected wavelengths using filters at 390±125 nm and >715 nm in snow-covered *Alnus incana* (gray alders) shrubs in the boreal forest near Laval University and on nearby grassland during a whole winter by placing light sensors at fixed heights in shrubs and on grassland. Irradiance in shrubs was greatly reduced at 390 nm and much less at 760 nm, where ice is much more absorbent. We performed radiative transfer simulations, testing the hypothesis that shrub branches behave as homogeneous absorbers such as soot. At 390 nm, dense shrub branches are found to reduce irradiance similarly to about 140 ppb of soot. For the >715 nm wavelengths, insufficient data and the greater ice absorption do not allow accurate conclusions. Noting that photochemically active radiation is mostly in the near UV and blue, we calculate that a high branch density will reduce photochemical reaction rates integrated over the whole snowpack by about a factor of two. This may affect the composition of the lower Arctic atmosphere in winter and spring in numerous ways, including a lower oxidative capacity, lower levels of nitrogen oxides and modified secondary aerosol production. Climatic effects are expected from these compositional changes.



## 1 Introduction

Quantifying the irradiance within the snowpack is of interest for numerous reasons, including metamorphism and photochemistry. Here, we focus on this latter aspect. Snowpack photochemistry modifies the snow composition and produces numerous reactive molecules such as NO and $NO_2$ (Honrath et al., 1999), formaldehyde (Sumner and Shepson, 1999), heavier carbonyl compounds and alcohols (Boudries et al., 2002; Houdier et al., 2002) carboxylic acids (Dibb and Arsenault, 2002) and haloalkanes (Swanson et al., 2002). The emission of these species to the atmosphere considerably affects the atmospheric oxidative capacity, ozone and aerosol formation (Finlayson-Pitts and Pitts, 1993; Domine and Shepson, 2002; Grannas et al., 2007), with potential climatic effects because ozone is an important greenhouse gas (Worden et al., 2008) and aerosols scatter light and act as could condensation nuclei (Farmer et al., 2015). Quantifying snowpack emissions to the atmosphere caused by photochemistry requires the knowledge of the irradiance profile in the snowpack (France et al., 2010).

Most snowpack photochemical reactions are triggered by radiation in the 300 to 450 nm wavelength range (Grannas et al., 2007; Wang, 2021), so that spectral data is required to investigate this topic. Vertical profiles of spectral irradiance have been measured in snow containing light-absorbing impurities in the form of deposited aerosols (King and Simpson, 2001; Simpson et al., 2002; France et al., 2011a; France et al., 2011b; Picard et al., 2016). In the Arctic, shrubs are expanding (Sturm et al., 2001b; Ropars and Boudreau, 2012; Ju and Masek, 2016) and the thin Arctic snowpack usually does not extend above the top of shrubs, especially in the high Arctic, where shrubs rarely exceed 50 cm in height (Sturm et al., 2001a; Marsh et al., 2010; Domine et al., 2016) so that buried shrub branches affect light propagation there. The interest of understanding light propagation in snow with shrub branches therefore increases as climate warms, for the climate-relevant reasons mentioned above. A few studies have been devoted to the measurement or modeling of the impact of shrubs on irradiance in the presence of snow (Bewley et al., 2007; Liston et al., 2007; Marsh et al., 2010). These studies however took place in the low Arctic and were focused on the extinction caused by shrubs protruding above the snow surface and their objective was related to the energy budget at the snow surface. Spectral data or extinction below the snow surface were not considered so that applications to photochemistry are not possible. Vertical spectral irradiance profiles have been measured by (Belke-Brea et al., 2021) in the low Arctic within shrubs and in snow without shrubs. The shrubs were dwarf birch (*Betula glandulosa*) whose branches are very supple and were bent far below the snow surface so that their effect on irradiance did not extend over the whole snowpack. The impact of shrubs on photochemistry thus cannot be simply evaluated in this case.

To contribute to the understanding of shrub effects on irradiance profiles in snow, we have monitored visible light fluxes at wavelengths centered on 390 nm (blue) and 760 nm (red) in snow with shrubs and in nearby sites without shrubs during the whole 2020-2021 winter. The 390 nm wavelength is within the most photoactive wavelength range and the ice optical properties do not vary much within this range (Picard et al., 2016) so that findings at 390 nm probably apply reasonably well to the whole photochemical range. At 760 nm, photochemistry is not known to be active for most molecules. However, at this



wavelength, the ice absorption coefficient is about 120 times greater than at 390 nm (Picard et al., 2016), so that investigating this longer wavelength informs us on the impact of shrubs under more absorbing ice conditions.

We deployed custom-made 390 and 760 nm light sensors at fixed heights at regular intervals before snow onset. We thus obtained irradiance data in snow with and without shrubs. Using the data and radiative transfer simulations, we were able to quantify the impact of shrubs on light propagation and on depth-integrated light fluxes in the snowpack.

## 2 Instruments and Methods

### 2.1 Optical sensors

Each optical sensor consists of a frosted spherical polystyrene sphere 7 mm in diameter (Cospheric). The sphere had been perforated to its center to insert two plastic optical fibers with 1500 µm core diameter and 15 µm-thick cladding (Toray Industries, PGR-FB1500). The two optical fibers were inserted into a stainless-steel tube of gauge 6G (Component Supply, HTX-06T) to provide rigidity and protect the interface of the fibers and the plastic sphere (Fig. 1). The tube was completely covered with teflon up to the base of the sphere to mimic the snow reflectivity and to prevent water from seeping into the tube and damaging the optical fiber. The assembly, consisting of the sphere, the fibers, and the tube, was then inserted into a PVC mount, and the fibers were routed to a junction box on the ground (Fig. 1).

Inside the junction box, each fiber end was connected to a photodiode (Vishay, TEMD5080X01) which was mounted on a custom-designed printed circuit board (PCB). Blue and red optical filters (Hoya B390 and SCHOTT RG715) were glued with UV light cure adhesive (Loca TP2500) directly to the photodiode sensors. The blue filter has a transmission band centered at 390 nm with a bandwidth of 125 nm. We refer to the radiation transmitted by this filter as 390 nm (or blue) radiation. The red longpass filter transmits light above 715 nm, up to the wavelength allowed by the photodiode, around 1000 nm. We refer to the radiation transmitted by this filter as 760 nm (or red) radiation because 760 nm is near the middle of the relevant range transmitted through snow at significant depths (Warren and Brandt, 2008) and this wavelength was found to be a valid "effective" wavelength for subsequent single-wavelength simulations. The signal received from the photodiodes was conditioned through an amplification and noise-filtering circuit before being transmitted to an Arduino Nano acquisition board. The schematics of the optical assembly is shown in Fig. 1.



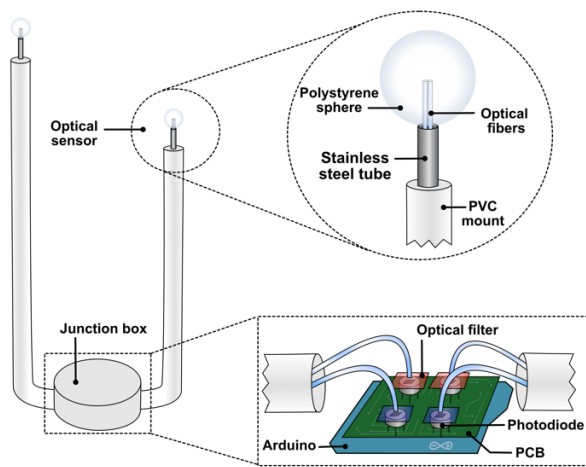

**Fig. 1. Schematic of optical sensors. Each assembly consists of two frosted spheres, each housing two optical fibers that are directed to a custom-made printed circuit board (PCB) located in the junction box. Photodiodes collect the signal from the fibers which is filtered by either a blue or red filter.**

## 2.2 Sensor deployment and site description

The sensors, assembled in pairs, were deployed in the Montmorency Forest, the research forest of Université Laval, 60 km north of Québec City. Ideally, an Arctic site would have been preferred. However, the logistical constraints linked to continuous monitoring and data transmission would have been enormous and the risk of instrument failure much greater. In any case, the Arctic was closed to travel at that time because of the COVID 19 pandemic, so that a nearby site was mandatory. The study site (47.3352°N, 71.1375°W, 678 m elevation) featured alder shrubs (*Alnus incana ssp. rugosa*) 1.5 to 2 m high next to a large clearing without erect vegetation, so that shading of the sun was minimal except for very high solar zenith angles (SZA). A set of four sensor pairs located at the tip of PVC mounts of heights 325, 485, 650, 850, 1000, 1200, 1375 and 1550 mm above the ground were placed in the clearing (hereafter F sensors, in the FIELD spot). The heights were selected based on usual peak snow heights at the site, about 2 m, based on the nearby NEIGE monitoring station 1.75 km to the SW at an elevation of 665 m (Pierre et al., 2019). A similar set of eight sensors was placed in four similar shrubs (hereafter S sensors, in the SHRUB spot), about 5 m to the north of the FIELD site (Fig. 2). Several striped poles in view of an automatic time lapse camera were placed in the shrubs and in the field to document snow height evolution. The NEIGE station is equipped with an ultrasonic snow height gauge and a CNR4 radiometer (Kipp & Zonen) which provides broadband (300-2800 nm) downwelling solar irradiance.

It is important to note that each SHRUB sensor had its own peculiar environment within shrub branches. The paired design also imposed constraints on placement. In particular, the 325 mm SHRUB sensor, called S325, was well in the middle of a shrub, with several large branches in its vicinity. The S485 and S650 sensors were closer to the edge of a shrub, with fewer





and smaller branches in their vicinity (Fig. 2). These observations are important for data analysis. Fig. S1 in the Supplement details the positions of the S325 and S485 sensors during installation.

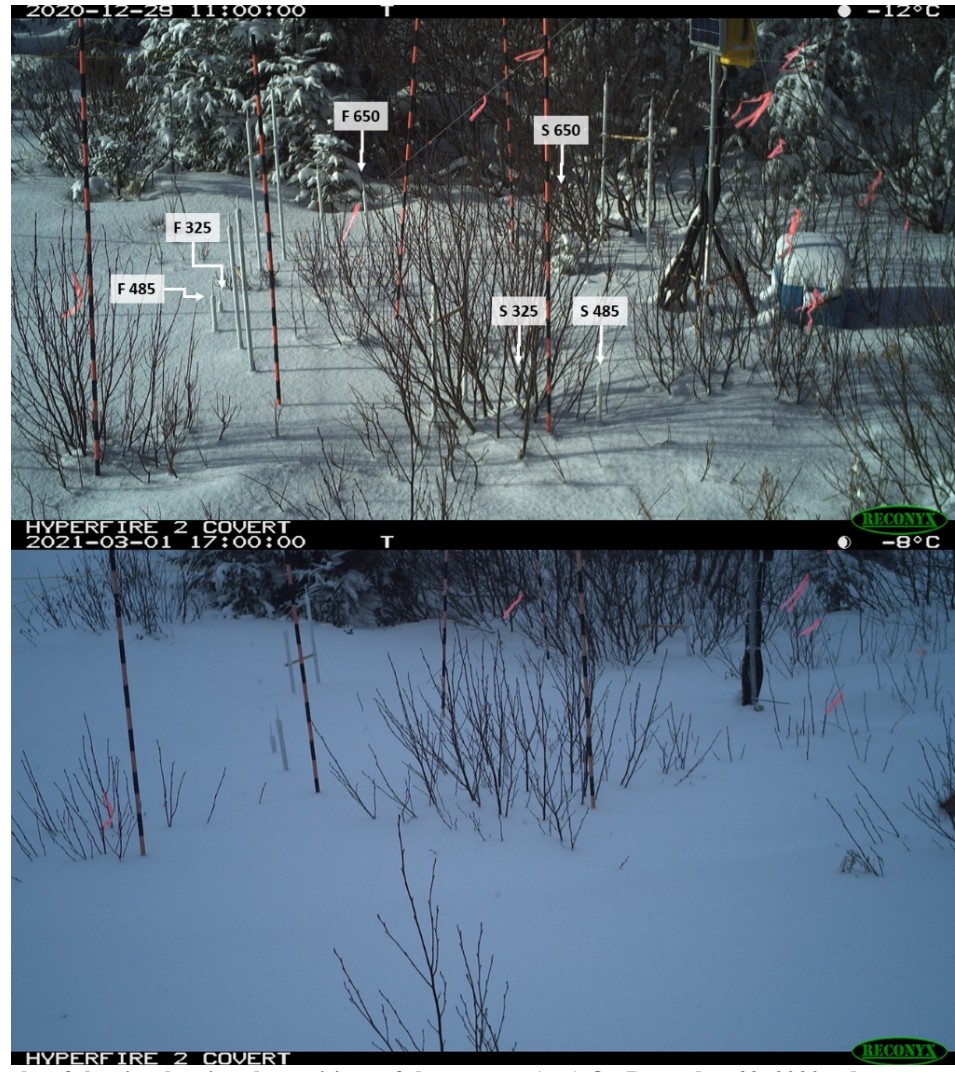

**Figure 2. Photographs of the site showing the positions of the sensors. (top) On December 29, 2020, when no sensor was covered by snow. The six sensors whose data were used here are indicated. S325, S485 and S650 are in the SHRUB spot at heights of 325, 485 and 650 mm. F325, F485 and F650 are in the FIELD site. (bottom) Same view at peak snow height on March 1st, 2021, when the six sensors were buried. Note that the snow bent the branches, modifying the branch environment of the S sensors visible on top.**

To help quantify branch characteristics, on November 2nd 2022, the morphology of a representative shrub was investigated by measuring the number and diameters of the branches at the heights where our data analysis was focused: 325, 485 and 650 mm. These data can also be used for comparisons with shrubs elsewhere in past and future studies, e.g., (Belke-Brea et al., 2020).



**2.3 Electronics, data acquisition and transmission**

The schematic of the electronics used for data acquisition and transmission is shown in Fig. 3. As the project involved the deployment of 16 optical sensors, a custom printed circuit board (PCB) was designed to provide a scalable, low-power, and cost-effective solution. Each PCB included four photodiodes (Vishay, TEMD5080X01), four operational amplifier (op-amp) circuits (STMicroelectronics, TSZ124) as well as an Arduino Nano for signal acquisition. The op-amp included a variable gain control on each channel, enabling the adjustment of the output response for each photodiode independently. This adjustment was necessary to ensure that the signal from each photodiode covered the entire 5V input range of the Arduino Nano analog-to-digital converter.

The Arduino Nano in each junction box was equipped with a DHT22 temperature sensor which showed that the temperature at the ground level remained constant near 0°C throughout the winter. No correction for the op-amp temperature-dependant response was thus required. The Arduino Nanos were also connected and powered through USB serial communication with a primary Raspberry Pi (RP1). Each Arduino continuously transmitted raw data from all four photodiodes, along with temperature and humidity readings. This RP1 was equipped with a Python acquisition script programmed to receive the data stream from each Arduino and to compute a single reading per sensor every 10 minutes. This strategy helped limit the size of the files which later needed to be transferred to the server over the internet.

A 3G communication module was added to transfer the data in real-time to a server. However, due to the high-power consumption of the module, a secondary Raspberry Pi (RP2) was used for powering and communicating with this 3G module, while the RP1 managed and powered the eight Arduino Nanos. The RP2 was also tasked to monitor the field with a Raspberry Pi camera (v.1.3) to provide an overall perspective of the study site. These images were transferred to the server in real-time.

To minimize the power consumption, a timer relay module (DS1307 real-time clock & SRD-12VDC-SL-C) controlled by an Arduino Nano was used to power both Raspberry Pis. The relay was programmed to keep the Pis powered on during the daytime hours (7AM to 6PM), with a 30-second power interruption every 10 minutes for rebooting and potential bug resolution.



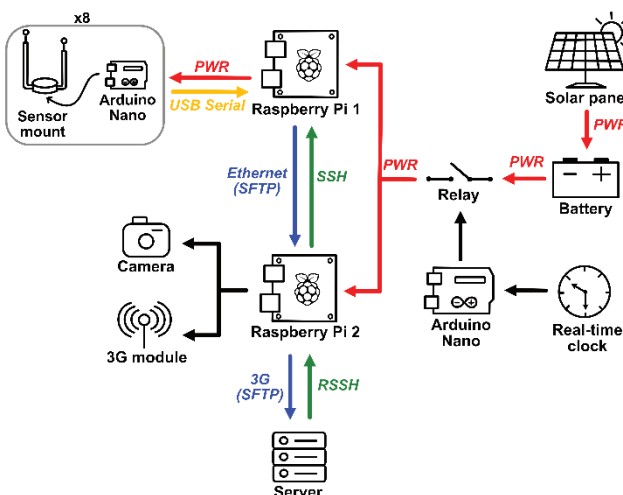

**Figure 3. Schematic of the signal acquisition and transmission system. The Raspberry Pi 1 received the data stream from each of the eight sensor pairs, while the Raspberry Pi 2 handled data transmission over internet to the server. The whole system was operated exclusively during daytime hours (7AM-6PM).**

## 2.4 Snow height determination

The snow height was determined from camera photos taken on site (four photos per day). The striped poles depicted in Fig. 1 served as reference units for image analysis. To improve the time resolution, the two daily measurements were supplemented with snow height data from the NEIGE station, which provided hourly measurements. Considering spatial variations, we estimate the uncertainty on snow height to be 1.5 cm.

## 2.5 Field measurement of snow properties

To constrain inputs into radiative transfer models, vertical profiles of snow properties were measured on January 6th, February 2nd, February 23rd and April 5th 2021. Profiles of density and specific surface area (SSA) were measured at sites near and similar to FIELD and SHRUB. Methods were similar to those described in (Domine et al., 2015). Briefly, a 100 cm$^3$ density cutter and a field scale were used for density. SSA was determined using the DUFISSS instrument based on the measurement of the infrared reflectance of snow at 1310 nm using an integrating sphere (Gallet et al., 2009).

## 2.6 Analysis of irradiance data

Our interest is in understanding the irradiance in the snow as a function of depth below the snow surface and of incident solar radiation. The variable of actual interest is the fraction of light incident on the snow surface that reaches the buried sensor. The signal from each sensor, $I_{r,i}$, was therefore normalized to the incident radiation $I_0$, which was obtained from sensors that were



not buried and whose signal was not saturated. For the blue radiation, we used the average of the $I_{r,i}$ of the three FIELD sensors at heights of 1000, 1200 and 1375 mm to obtain $I_{0b}$. For the red radiation, the signal from these three FIELD sensors was saturated, so we used the SHRUB sensor at a height of 1550 mm to obtain $I_{0r}$. There were slight gain variations among sensors.

The analysis of data when no sensor was buried and there was just a few cm of snow on the ground in December allowed the homogenization of all the gains. The gain correction factors range from 0.85 to 1.35. For comparisons with simulations, we therefore used gain-corrected data, $(I_{r,i}/I_0)_c$. For the SHRUB sensors, $(I_{r,i}/I_0)_c$ therefore takes into account the shading by shrub branches. The detection limit for $I_{r,i}/I_0$ was found to be 0.002.

Under blue sky conditions, most of incident radiation is direct so that the angular response of the sensors used as references

must be known. This angular response was measured in the laboratory using a goniometer and was found to be different from a cosine response. Our laboratory measurements allowed correcting for this angular response. However, it was simpler and probably less error-prone to limit our data analysis to overcast conditions, when incident light was diffuse, similar to the conditions of the sensors buried in the snow. Subsequently, we therefore only consider $(I_{r,i}/I_0)_c$ without angular correction, on overcast days. Limiting analysis to overcast conditions is not expected to affect our general conclusions because in snow, light

becomes diffuse after a depth of generally a few mm (Simpson et al., 2002). Since our irradiance measurements are all below this depth, the nature of the light incident at the surface (direct vs diffuse) has no impact on our irradiance analysis. Overcast conditions were determined from time lapse images and from irradiance values. The normalized irradiance values were obtained by averaging data over several hours when overcast conditions were certain. We estimate the uncertainty on $(I_{r,i}/I_0)_c$ to be 15%, except for values <0.005, when it is about 80%.

The irradiance profiles were simulated using the 'TARTES' Python module (Libois et al., 2013), available at (https://snow.univ-grenoble-alpes.fr/tartes/ last accessed on May 23$^{rd}$ 2024). In TARTES, input data are the thickness, density and SSA of each snow layer. Additionally, absorption by impurities is considered. Here, we consider for simplicity that all absorbing impurities are soot, with properties reported in (Bond and Bergstrom, 2006), i.e., the soot density is 1800 kg m$^{-3}$ and its complex optical constant is 1.95-0.79i, independent of wavelength. To analyze the effect of shrub branches with TARTES,

we consider the soot-equivalent of branches to simulate their absorption. This approach therefore makes the hypothesis that branches behave as a homogeneous absorber such as soot, even though they are discrete absorbers. With this approach, we determine the soot-equivalent of shrub branches by comparing the concentrations of absorbing impurities between the FIELD and SHRUB simulations. At 390 nm, we calculate that for typical snow encountered during this study (density=200 kg m$^{-3}$, SSA=25 m$^2$ kg$^{-1}$, soot=25 ng g$^{-1}$), irradiance is reduced by a factor of 10 over a distance of 29 cm. The sensors will therefore

be sensitive to optical properties within a radius of ~29 cm.

At 760 nm, ice absorbs 62 times more than at 390 nm (Warren and Brandt, 2008; Picard et al., 2016), as shown in Fig. 4. Soot absorbs 0.51 times as much because of the inverse wavelength dependence of absorption on the imaginary optical index. Regarding bark optical properties, (Juola et al., 2022b) report average reflectance values of several *Alnus incana* individuals and also provide numerical data between 397 and 1000 nm (Juola et al., 2022a). The reflectance is 0.18 at 397 nm and 0.53 at




760 nm (Figure 4). In absorbtivity terms, assuming bark is thick enough, this yields values of 0.82 at 390 nm and 0.47 at 760 nm, a decrease of a factor of 0.57, not very different to the soot value.

Because of the greater ice absorption, at 760 nm irradiance in snow decreases by a factor of 10 over a distance of 8.2 cm, much shorter than at 390 nm. The corresponding volume is 44 times smaller than at 390 nm. The sensor will therefore be more sensitive to its immediate environment, such as the proximity of a thick branch.

Even though considering irradiance reduction by a factor of 10 is intuitively simple, the e-folding depth is often considered preferably (Simpson et al., 2002; France et al., 2010; France et al., 2011a). This is the depth over which irradiance is reduced by a factor $1/e=0.368$. In the above case, the e-folding depths are 12.5 and 3.7 cm at 390 and 760 nm, respectively. We subsequently consider e-folding depth to quantify light extinction in snow.

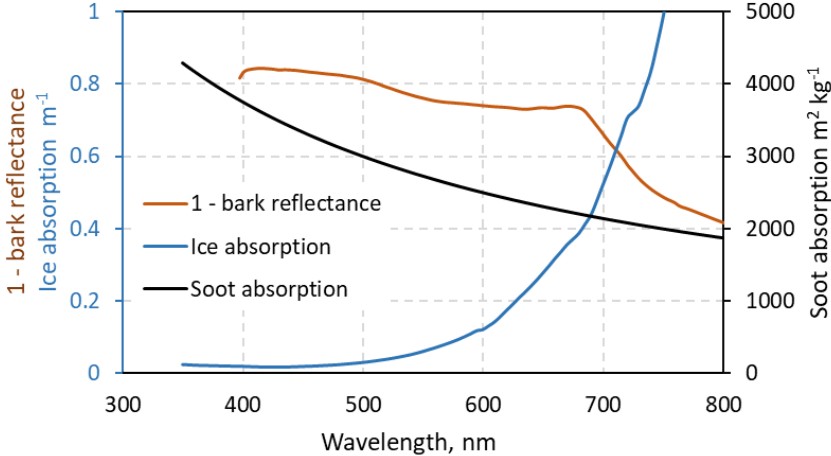

**Figure 4. Absorption coefficients of ice and soot used in this study. The reflectance of *Alnus incana* bark provided by (Juola et al., 2022a) is also shown.**

## 3 Results

### 3.1 Snow properties

Vertical profiles of snow density and SSA measured on February 23$^{rd}$ 2021 at spots similar to FIELD and SHRUB and within
50 m of those spots are shown in Fig. 5. Similar snow data were obtained on January 6$^{th}$, February 2$^{nd}$ and April 5$^{th}$. The data obtained on these last two dates are shown in the Supplement, Figs. S2 and S3. The data obtained on January 6$^{th}$ were not used since no sensor was covered at that time and no simulations were performed for that date.



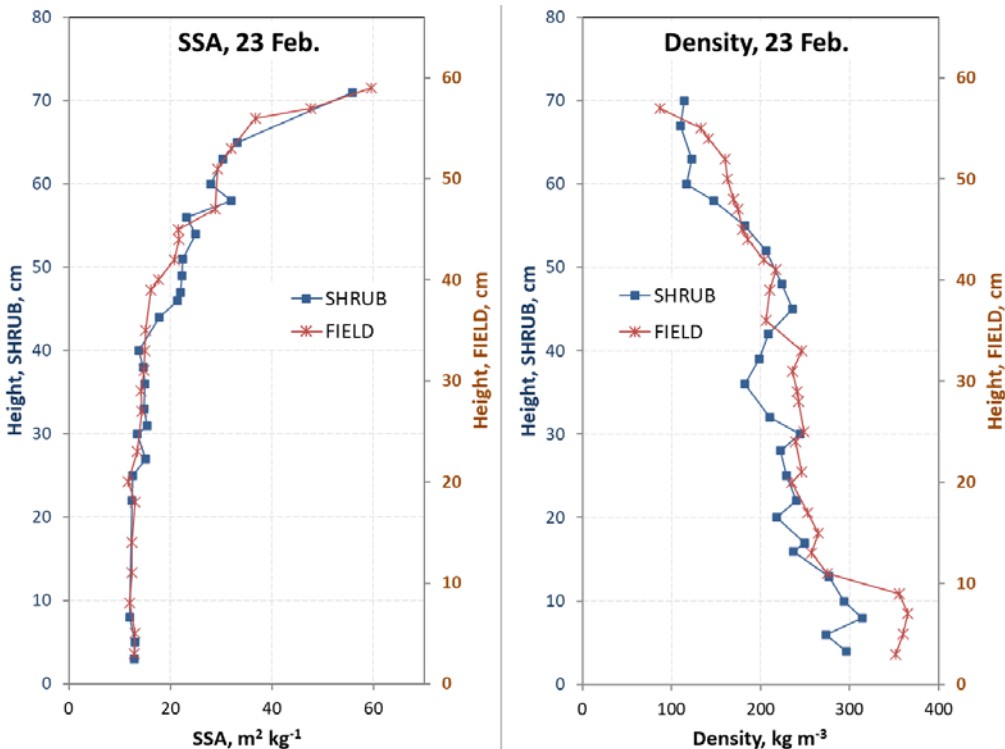

**Figure 5. Vertical profiles of snow specific surface area (SSA) and density measured at an open spot similar to FIELD and at a spot in shrubs similar to SHRUB on February 23rd 2021.**

The 2020-2021 winter was unusual in that the peak snow height was less than half the usual value (Bouchard et al., 2023) so that most of our sensors were not covered. Time series of snow height, as well as dates selected to perform irradiance simulations, are shown in Fig. 6.

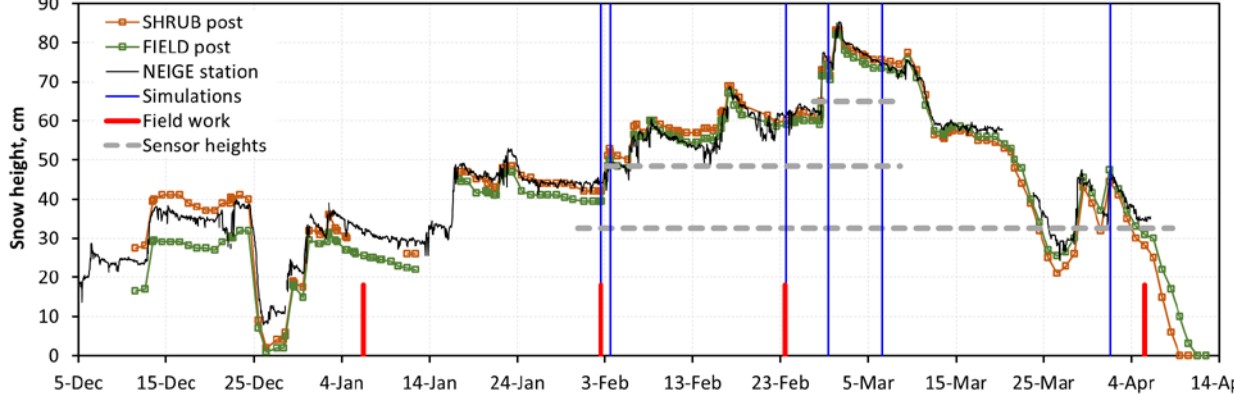

**Figure 6. Time series of snow height at the striped poles placed at the study site and determined daily from time lapse images. Snow height monitored by an ultrasonic gauge at the NEIGE station 1.75 km away is also shown. Dates of field work when snow density and specific surface area were measured are indicated with bold red vertical bars. Dates for which irradiance profiles were simulated**





**are noted by thin blue vertical lines. The heights of the three sensors that provided irradiance data have been reported with horizontal dashed lines on the graph.**

**3.2 Irradiance data**

Fig. 7 shows plots of normalized irradiance data $I_{r,i}/I_{0b}$ for i=325, 485 and 650 mm, for both FIELD and SHRUB during the February $28^{th}$ to March $7^{th}$ period. This was the only time period when sensors at three heights were buried. The downwelling shortwave irradiance measured by the CNR4 is also shown, which reveals that March $3^{rd}$ had perfect clear sky whereas March $2^{nd}$ had clear sky with just a few clouds. The reduced solar radiation on February $28^{th}$ and time-lapse images show this day was

totally overcast. On March $6^{th}$, conditions were mostly overcast with rather thin clouds and occasional breaks in the clouds. Both these days were chosen to perform radiative transfer simulations.

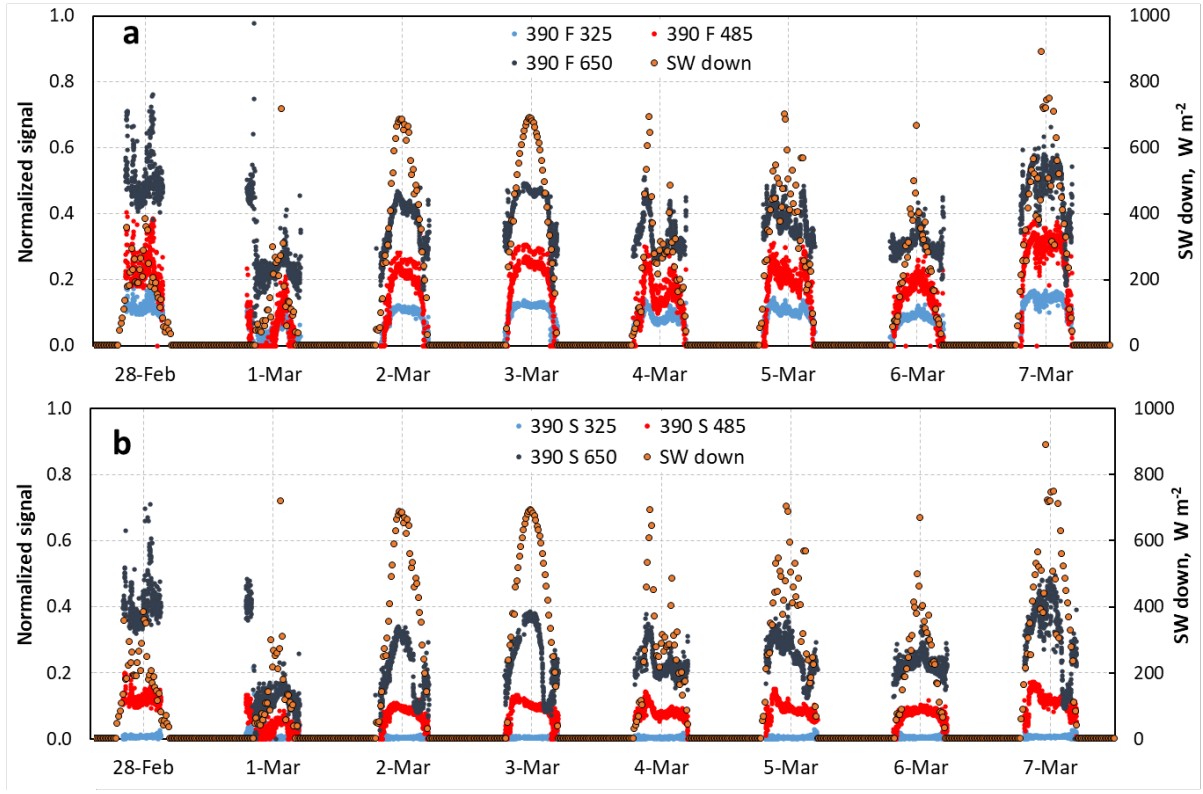

**Figure 7. Time series of normalized irradiance $I_{r,i}/I_{0b}$ at 390 nm for the three sensors at 325, 485 and 650 mm heights, when these three sensors were buried. The downwelling solar irradiance (SW down) from the CNR4 is also shown. (a) FIELD, (b) SHRUB. 390**
**F 325 is the signal for the 390 nm (blue) wavelength of the FIELD sensor at the 325 mm level. 390 S 325 is the same, but for the SHRUB sensor. SW down is the downwelling shortwave radiation measured by the CNR4.**





The signals during both days were determined by selecting periods when conditions were overcast with thick clouds, as shown in Fig. 8. For example, on February 28[th] at FIELD (Fig. 8a), the signal $I_{r,i}/I_{0b}$ was determined based on data between 10:30 and 12:30. As expected, irradiance signals are lower at SHRUB than at FIELD because of light absorption by shrub branches.

The other days selected for simulations were February 2[nd], 3[rd] and 23[rd] and April 1[st]. Only one to two sensors were then buried, as visible in Fig. 6. The normalized and gain-corrected experimental irradiance data $(I_{r,i}/I_{0b})_c$ and $(I_{r,i}/I_{0r})_c$ used are reported in the Supplement, Tables S1 and S2.

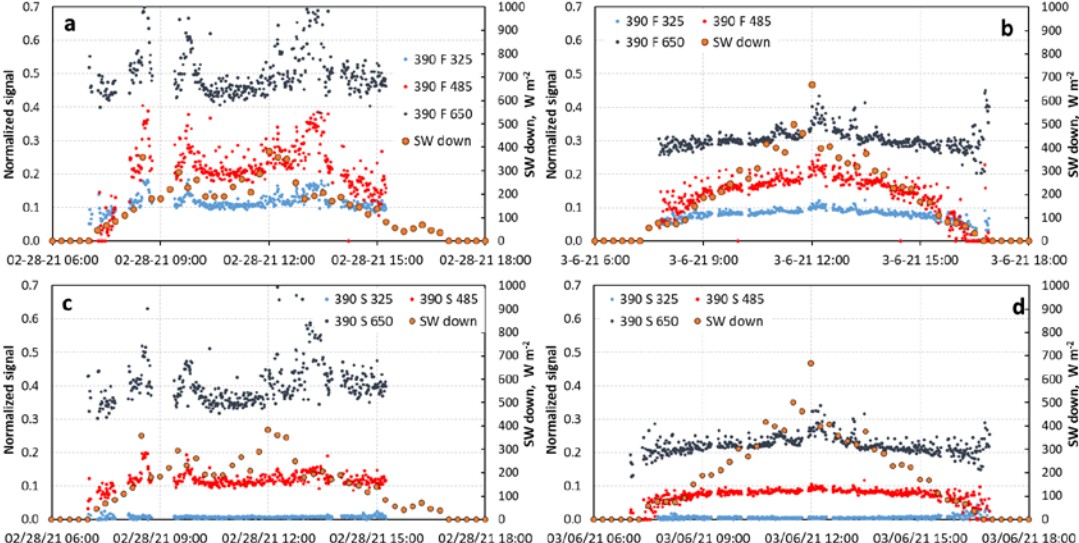

**Figure 8. Daily evolution of the normalized irradiance $I_{r,i}/I_{0b}$ under mostly overcast conditions. (a) FIELD, February 28[th] (b)**
**FIELD, March 6[th] (c) SHRUB, February 28[th] (d) SHRUB, March 6[th].**

Because of the greater ice absorption at 760 nm than at 390 nm (Warren and Brandt, 2008; Picard et al., 2016), radiation penetration is much shallower for red than for blue wavelength and a signal was detected only for the topmost sensor at the red wavelength. $I_{r,i}/I_{0r}$ data at 760 nm obtained from February 28[th] to March 7[th] are shown in Fig. 9.




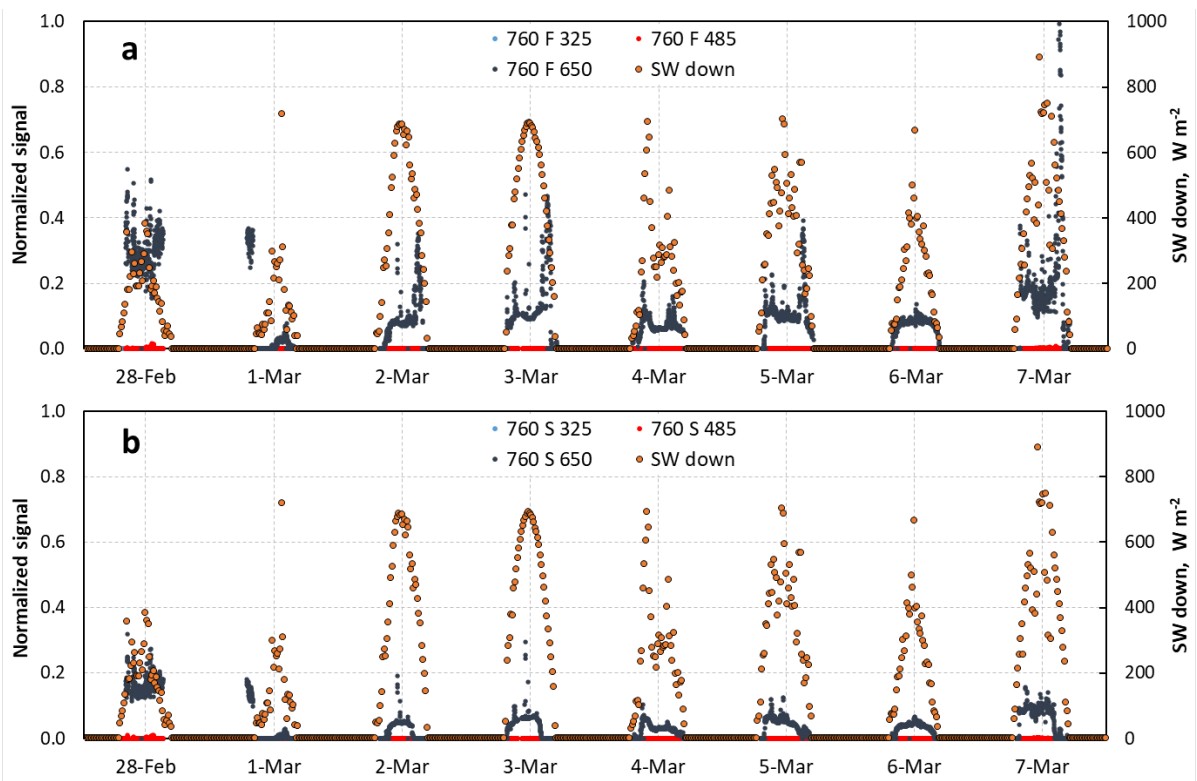

**Figure 9. Time series of normalized irradiance $\mathbf{I_{r,i}/I_{0r}}$ at 760 nm for the sensors at 325, 485 and 650 mm, when all three were buried. The downwelling solar irradiance (SW down) from the CNR4 is also shown. (a) FIELD, (b) SHRUB. The signals from the sensors at 325 and 485 mm are too low for detection.**

### 3.3 Shrub morphology

The mean diameter and number of branches of the shrub canopy in a representative shrub at the level of the S325, S485 and S650 sensors are reported in Table 1. The distribution of branch diameters at these same levels is shown in Fig. S4. The average canopy diameters at those heights were 75, 110 and 135 cm.

**Table 1. Canopy and branch characteristics at the level of the lowermost three sensors placed at SHRUB.**

| Sensor | Canopy diameter, cm | Number of branches | Mean diameter of branches, mm | Standard deviation of diameter, mm |
|---|---|---|---|---|
| S325 | 98 | 19 | 14.2 | 5.3 |
| S485 | 144 | 51 | 7.1 | 5.4 |
| S650 | 165 | 75 | 5.2 | 4.2 |



**3.4 Radiative transfer simulations**

Irradiance profiles were simulated on February 2nd, 3rd, 23rd and 28th, March 6th and April 1st. An irradiance profile can be simulated if the physical properties (SSA and density) and impurity concentrations of the snow layers are known. For February 2nd and 23rd, we used the physical data obtained on those very days during our snowpit measurements. For the other days, snow physical properties were estimated from the snowpit data, the literature (Domine et al., 2007; Taillandier et al., 2007) which helps in estimating the time-evolution of snow physical properties and the SSA-density correlation, and above all from our experience of snow physical properties and their evolution at the Montmorency Forest (Bouchard et al., 2022; Bouchard et al., 2023). The concentration of impurities in the snow, treated as soot-equivalent, was not measured and was used as an adjustable variable to optimize the agreement between measured and simulated irradiance profiles. The main objective of these simulations is to compare the impurity concentrations between FIELD and SHRUB in order to deduce a soot-equivalent for the absorption by shrub branches.

Physical and chemical snow variables used for simulations at 390 nm are reported in Table 2. Layer numbers and boundaries were determined so that density and SSA showed limited variations within a layer (see Figs. 4, S2 and S3). Soot concentrations were determined for each layer on each simulation day. However, the soot concentration of a given layer is not expected to vary significantly over time. Adjusted soot concentrations were therefore slightly modified to limit variations in a given layer over time. Variations can indeed occur because of experimental error and can also occur naturally. For example, recrystallization during metamorphism may free an impurity particle embedded into a crystal, so that its location is then on the surface of a crystal, where its absorbing effect is reduced (Kokhanovsky, 2013). Freed particles may also drop to a lower level. The snow physical properties at the sensors' location are also probably slightly different to those at the measurement spots. The values of Table 2 allow a reasonable agreement between simulations and data at 390 nm, as shown in Fig. 10, except on April 1st. Using lower soot values on that last date would allow a perfect fit, but decreasing soot values during melt would not make physical sense. We reflect on this situation in the discussion.





**Table 2. Snow density, specific surface area and soot profile values used in radiative transfer simulations using the TARTES model at 390 nm. Snow layers are listed from the top down. Soot values are in part per billion by mass (ppb, ng g⁻¹).**

| FIELD | | | | SHRUB | | | |
|---|---|---|---|---|---|---|---|
| Layer thickness, m | Layer density, kg m⁻³ | Layer SSA, m² kg⁻¹ | Soot, ppb | Layer thickness, m | Layer density, kg m⁻³ | Layer SSA, m² kg⁻¹ | Soot, ppb |
| **February 2ⁿᵈ** | | | | | | | |
| 0.02 | 180 | 34 | 25 | 0.02 | 150 | 45 | 150 |
| 0.1 | 210 | 22 | 25 | 0.1 | 180 | 27 | 150 |
| 0.1 | 200 | 17 | 25 | 0.1 | 210 | 22 | 170 |
| 0.18 | 250 | 15 | 30 | 0.2 | 240 | 15 | 170 |
| **February 3ʳᵈ** | | | | | | | |
| 0.1 | 110 | 75 | 15 | 0.1 | 110 | 75 | 100 |
| 0.02 | 180 | 33 | 25 | 0.02 | 180 | 33 | 100 |
| 0.1 | 210 | 21 | 25 | 0.1 | 200 | 21 | 110 |
| 0.1 | 200 | 17 | 25 | 0.1 | 200 | 17 | 160 |
| 0.18 | 250 | 15 | 30 | 0.19 | 250 | 15 | 170 |
| **February 23ʳᵈ** | | | | | | | |
| 0.1 | 140 | 55 | 50 | 0.1 | 120 | 50 | 60 |
| 0.1 | 180 | 25 | 10 | 0.1 | 200 | 25 | 50 |
| 0.3 | 210 | 15 | 20 | 0.3 | 200 | 15 | 150 |
| 0.1 | 350 | 13 | 30 | 0.11 | 290 | 13 | 170 |
| **February 28ᵗʰ** | | | | | | | |
| 0.125 | 100 | 60 | 20 | 0.125 | 90 | 60 | 40 |
| 0.1 | 140 | 40 | 40 | 0.1 | 120 | 40 | 60 |
| 0.1 | 180 | 25 | 10 | 0.1 | 200 | 25 | 30 |
| 0.3 | 210 | 14 | 13 | 0.3 | 210 | 14 | 150 |
| 0.1 | 350 | 13 | 28 | 0.12 | 300 | 13 | 170 |
| **March 6ᵗʰ** | | | | | | | |
| 0.075 | 150 | 55 | 25 | 0.075 | 145 | 58 | 60 |
| 0.07 | 150 | 50 | 25 | 0.07 | 145 | 52 | 50 |
| 0.09 | 155 | 37 | 17 | 0.09 | 150 | 36 | 25 |
| 0.1 | 180 | 25 | 10 | 0.1 | 180 | 25 | 25 |
| 0.3 | 215 | 14 | 11 | 0.3 | 215 | 14 | 150 |
| 0.1 | 360 | 13 | 30 | 0.12 | 310 | 13 | 170 |
| **April 1ˢᵗ** | | | | | | | |
| 0.11 | 120 | 37 | 20 | 0.11 | 110 | 37 | 100 |
| 0.1 | 400 | 11 | 20 | 0.1 | 320 | 11 | 100 |
| 0.25 | 420 | 5 | 30 | 0.22 | 360 | 5 | 170 |



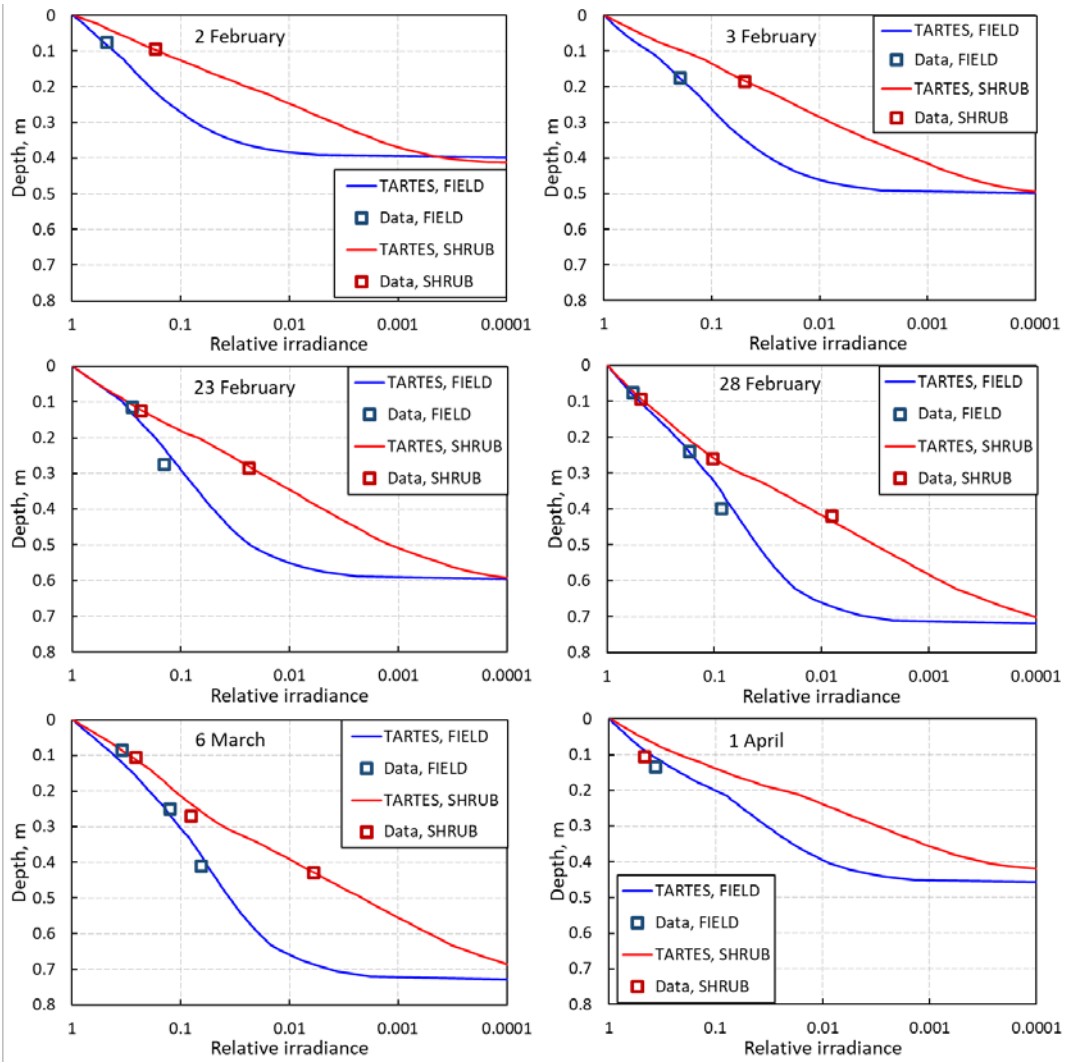

**Figure 10. Data and simulations of irradiance profiles at FIELD and SHRUB for 390 nm. The scale of the Y axis varies because snow depth varied. One to three data points were obtained, depending on the number of sensors buried. The X-scale has been inverted to start at the snow surface. The size of the data points represent the error bars.**

For the red radiation, our filter transmits all wavelengths >715 nm. In this range, the absorption coefficient increases with increasing wavelength (Warren and Brandt, 2008): it is 0.52 m$^{-1}$ at 700 nm and 2.1 m$^{-1}$ at 800 nm. Performing simulations over a wavelength range that accounts for dispersion is difficult without knowing the incident spectral irradiance and we rather took the option of choosing an effective wavelength. Simulations with TARTES showed that for wavelengths >850 nm, the signal at 10 cm depth, typical of the position of the topmost sensor, was negligible. The incident irradiance is also depleted in those wavelengths under cloudy conditions. TARTES simulations using a range of snow physical and chemical properties indicated





that 760 nm was a suitable estimate of an effective wavelength for the conditions encountered. TARTES simulations were therefore performed using the 760 nm wavelength optical properties of ice and soot.

For the 760 nm simulations, the same snow physical properties were used as for the 390 nm wavelength. Regarding impurity
concentrations, the absorption spectrum of the impurities present here may be different from that of (Bond and Bergstrom, 2006), so that a constant multiplicative factor may have to be used to adjust all soot concentrations at 760 nm. For FIELD data, it was found that this factor was 1, suggesting that most impurities present in the snow were probably soot with optical properties similar to those of (Bond and Bergstrom, 2006). Fig. 4 indicates that bark absorptivity decreases by a factor of 0.57 between 397 and 760 nm, not very different to the factor for soot in this same wavelength range: 0.53. Given the variability in
bark properties of individual trees however (Juola et al., 2022b), the bark factor may be different in our specific case.



**Figure 11. Data and simulations of irradiance profiles at FIELD and SHRUB for 760 nm. Only the topmost sensor produced a detectable signal. Sensors levels in mm are indicated in the legend of each panel. For SHRUB, simulations with impurity concentrations multiplied by 0.33 and 3 are also shown.**

We therefore performed simulations at 760 nm using the same soot concentrations for SHRUB as at 390 nm. To investigate the impact of bark optical properties that would vary with wavelength differently to soot, we also performed simulations with concentration multiplied by 0.33 and by 3. Results of simulations at 760 nm are shown in Fig. 10, for sensors for which there was a detectable signal. For the February 3$^{rd}$ data, the 325 mm sensor was 185 mm below the snow surface, and its signal was below detection limit. The corresponding curve is therefore not shown.


## 4 Discussion

### 4.1 Snow modification by melting on April 1$^{st}$

The April 1$^{st}$ data do not show any effect of shrubs branches on irradiance and simulations do not agree with data, while the agreement is good for all other dates. We hypothesize that the intense melting event that occurred between March 9$^{th}$ and 26$^{th}$

and on March 31$^{st}$ (see the rapid snow height decreases in Fig. 6) led to the formation of percolation channels that perturbed radiative transfer differently at FIELD and SHRUB. These channels form ice fingers made up of highly clustered large grains (see e.g. Fig. 6 of (Bouchard et al., 2024)) which scatter light much less than the snow used in the simulations, therefore increasing irradiance at depth. Radiative transfer then cannot be simulated using plane-parallel snow layer geometry as done here. We therefore exclude the April 1$^{st}$ data from subsequent analysis.

### 4.2 Soot equivalent of shrub branches at 390 nm

The extra extinction at SHRUB relative to FIELD is mostly due to absorption by branches, and to a much lesser extent by different snow physical properties. By subtracting the FIELD soot values to those of SHRUB in Table 2, it appears that the soot equivalent of shrub branches at 390 nm is 125 to 145 ppb for the lower 40 cm of the snowpack. It is 10 to 85 ppb for the upper layers, with most values in the lower part of this range. This is consistent with the positions of the sensors. The 325 mm

sensor is in the middle of a shrub and is affected by a greater number of branches than both upper sensors, which are closer to the edge of a shrub as illustrated in Fig. 1 and Fig. S1.



### 4.3 Soot equivalent of shrub branches at 760 nm

It is noteworthy that the 760 nm simulations for FIELD reproduce the data well using the values used at 390 nm without any adjustment. This supports the validity of our approach and the choice of the 760 nm wavelength as an effective wavelength.

At SHRUB, Fig. 11shows that on February 23$^{rd}$ and 28$^{th}$ and on March 6$^{th}$, branches affect irradiance profiles in a negligible manner and changing the impurity concentrations by a factor of 3 has a negligible impact on irradiance at the sensors' depths. Data from those dates are from the S485 and S650 sensors, which are in low branch density spots. Given the short e-folding depth at 760 nm, the sensors are not affected by branches further away. Since ice is a significant absorber at 760 nm, the main absorber under these conditions is ice and it is therefore not possible to quantify precisely the weak extinction caused by branches. On February 2$^{nd}$ on the contrary, the S325 sensor is in a high branch density area and branch impact on the absorption profile is clearly visible. The fit that best reproduces the data is that with a soot concentration multiplied by 3 (Fig. 10). This could be interpreted by a stronger absorption of bark at 760 nm than at 390 nm, contrary to the data of Fig. 4. An alternative explanation is to note that, as shown in Fig. S1, the S325 sensor has several branches within an e-folding depth at 390 nm (12.7 cm), so that here branches have an important contribution to extinction. At 760 nm, the short e-folding depth (3.7 cm) implies that the signal is very sensitive to the immediate environment. It is likely that the extinction must be interpreted in terms of nearby branch density rather than in terms of bark optical properties. In other words, while at 390 nm data appear consistent with our hypothesis that branches can be treated as a homogeneous absorber with well-defined optical properties because the sensor then probes a large volume, this does not seem to be the case at 760 nm where the immediate proximity of branches determines extinction more strongly than bark optical properties.

### 4.4 Impact of shrubs on irradiance in snow

Figs. 9 and 10 illustrate that irradiance decreases faster with depth at SHRUB than at FIELD. This effect is mostly due to the presence of branches but also, to a lesser extent, to different snow physical properties. To isolate the contribution of branches, we calculate irradiance profiles at SHRUB with the same snow physical properties as at FIELD, but with the SHRUB equivalent-soot concentration. This allows us to determine the extinction caused by branches at 15 cm depth. The extinction factor for the case of high branch density was calculated with the February 2$^{nd}$ data, while the case of low branch density was calculated with the February 28$^{th}$ data. Results are summed up in Table 3. The 760 nm factors were calculated assuming that the spectral dependence of bark absorption was similar to that of soot. Table 3 shows that the radiative impact of a low branch density is very limited, while a high branch density reduces irradiance at 15 cm depth by a factor a 4.33 at 390 nm, a high value.



**Table 3. Radiation attenuation factor caused in snow by branches of low (Feb. 28th data) and high (Feb. 2nd data) densities at 15 cm depth. The radiation e-folding depth for the snowpack of February 2nd is also indicated.**

|  | 390 nm | 760 nm |
|---|---|---|
| High branch density | 4.33 | 1.42 |
| Low branch density | 1.32 | 1.05 |
| e-folding depth, cm | 11.1 | 3.4 |

Based on photographs of the S325 and S485 sensors taken during installation (Fig. S1) and also on the data of Fig. S4 in the Supplement, we attempt to estimate the number and mean diameter of branches within two e-folding depths of each sensor, for both wavelengths studied. These estimates are reported in Table 4. A strict proportionality between branch surface area within the distance of two e-folding depths and the attenuation factors of Table 3 is not expected because a full analysis would require 3-D radiative transfer modeling. However, it is clear that there is a reasonable correlation between Tables 3 and 4. We

can even tentatively estimate that 4 branches of 15 mm diameter within a distance of 22 cm from the sensor is roughly equivalent to 140 ppb of soot for the 390 nm radiation. The 760 nm data show that branches only have a local effect and that only high branch densities will detectably affect 760 nm irradiance in snow.

**Table 4. Number of branches and their mean diameter within a radius of two e-folding depths for the S325 and S485 sensors, for both wavelengths studied.**

|  | 390 nm | | 760 nm | |
|---|---|---|---|---|
| Sensor | No. of branches | Mean diameter | No. of branches | Mean diameter |
| S325 | 4 | 15 mm | 1 | 15 mm |
| S485 | 3 | 8 mm | 1 | 6 mm |


### 4.5 Impact of branches on snowpack photochemistry

In the Arctic, the snow height usually does not extend beyond shrub height, as often observed (Domine et al., 2016; Lafleur and Humphreys, 2018) and illustrated in Fig. S5 in the Supplement, because the trapping of wind-blown snow stops when most branches are buried. Therefore, the incident flux onto the snow surface is similar on snow with and without shrubs.

Photochemical reaction rates in any medium depend on irradiance. For species of atmospheric interest, the most active spectral region is the near UV where many molecules have absorption bands (Finlayson-Pitts and Pitts, 2000). The ice absorption coefficient varies little between 350 and 450 nm (Picard et al., 2016) so that the data obtained here at 390 nm may be used to obtain a first estimate of the impact of shrub branches on photochemical reaction rates. By integrating the irradiance over the depth of the snowpacks, the ratios of the rates SHRUB/FIELD have been calculated for the various snowpacks studied here.





On this case, we used the different snowpack physical properties for FIELD and SHRUB of Table 2, because this difference
is caused by the presence of shrubs (Domine et al., 2016). Results are shown in Table 4.

**Table 4. SHRUB/FIELD photolysis ratio in the snowpacks studied at five dates. Similar incident flux values are used for SHRUB and FIELD.**

| Date | 2 Feb. | 3 Feb. | 23 Feb. | 28 Feb. | 6 Mar. |
|---|---|---|---|---|---|
| Photolysis rate ratio | 0.47 | 0.57 | 0.75 | 0.79 | 0.68 |

For low branch density, typical of the environment of the top sensors on February 28[th], the integrated flux is reduced by just
21%. For high branch density (February 2[nd]), the flux is reduced by a factor greater than 2. These estimates assume that
branches have a radiative effect similar to homogenous impurities such as soot, an assumption that requires confirmation. In
the high Arctic (Domine et al., 2022) and even in the low Arctic (Belke-Brea et al., 2020) shrubs are comparatively shorter
than in this study and have thinner but more numerous branches. Furthermore, shrubs there often form dense bushes whose
branches overlap, increasing branch densities while in this study, shrubs were isolated. It is therefore likely that the effect of
branches on irradiance evidenced here would be at least at important, possibly more so, in the Arctic. We therefore speculate
that photolysis rates in shrub-covered areas in the Arctic may be reduced by at least 25 to 50%, as in Table 4, and possibly
even 75%. Furthermore, since branches are thinner and more numerous, the soot-equivalent approach used here, based on a
homogeneous medium approximation, may have an improved validity. We thus suggest that irradiance in shrubs in the Arctic
may be less spatially-variable than observed here and that a simple large-scale treatment may be valid.

The composition of the atmospheric boundary layer in polar regions is strongly affected by snowpack photochemistry. For
example, concentrations of nitrogen oxides are up to 100 times greater than expected by models that neglect snowpack
photochemistry (Honrath et al., 1999; Davis et al., 2001), because they are produced by the photolysis of the nitrate ion
contained in the snowpack. These high nitrogen oxides concentrations, together with snowpack photolysis of $H_2O_2$, (France et
al., 2007), increase atmospheric concentrations of the OH radical (Mauldin et al., 2004), which is the main atmospheric oxidant.
Decreased integrated irradiance fluxes in the snowpack caused by shrub expansion therefore has the potential to generate major
perturbations in the photochemistry of Arctic regions, in particular by increasing the lifetime of numerous species such as
organic molecules. Furthermore, changes in this chemistry may impact ozone concentrations, which are strongly affected by
the concentrations of both nitrogen oxides and organic compounds (Finlayson-Pitts and Pitts, 1993). Ozone is both a potent
greenhouse gas and a strong oxidant, so that the impact of shrub growth on atmospheric gaseous composition is manyfold and
complex, with processes contributing to ozone decrease and others to its increase. Besides gaseous concentrations, reduced
snowpack photochemistry is also expected to affect the formation of secondary organic aerosols from the atmospheric
oxidation of organic compounds (Hallquist et al., 2009). This will impact the number of cloud condensation nuclei (Farmer et
al., 2015), cloud droplet size and thus cloud albedo (Twomey, 1977). The reduction of snow photochemical rates by shrub
expansion may thus lead to numerous chemical and climatic effects that may deserve further quantification.

## 5 Conclusion

The comparative measurements of irradiance in snow with and without shrub branches show that when a high branch density is present, shrub branches produce an irradiance attenuating effect equivalent to about 140 ppb of soot at 390 nm, if the soot has optical properties similar to those mentioned by (Bond and Bergstrom, 2006). Such soot concentrations are those of

moderately to highly polluted snow (Wang et al., 2013; Dang et al., 2017). This reduces the rate of snowpack photochemistry by about a factor of 2, with the assumption that branches behave as a homogeneous impurity. At 760 nm, the impact of shrub branches is weaker because ice is then the main absorber. Since ice absorption increases regularly and monotonically between 450 and 760 nm (Warren and Brandt, 2008), we expect the impact of branches to decrease likewise over this spectral range.

### Supplement

The supplement contains Tables S1 and S2 and Figures S1 to S5.

### Code and data availability

The data used in the simulations have been reported in Table 1, Table S1 and Table S2. The TARTES code is available at
https://snow.univ-grenoble-alpes.fr/snowtartes/index.

### Author contribution

FD and MQ designed research with input from GP. DCC and FD obtained funding. MQ, LB, BB, SJ, RL, FLD and MAV built the experimental apparatus. GP wrote the radiative transfer simulation code. BB and FLD performed the snow field measurements. SOP measured the sensors optical properties. VD performed preliminary data analysis. FD and MQ analyzed the data and wrote the paper with input from GP. LB, BB, RL, FLD, MAV and DCC commented on the paper.

### Competing interest

Florent Domine is a member of the editorial board of The Cryosphere.

### Acknowledgements

This research was supported by the Sentinel North program of Université Laval, made possible, in part, thanks to funding from the Canada First Research Excellence. Marie-France Gévry managed the Sentinel North support and provided continuous



encouragements. Additional funding was provided by the NSERC-CREATE program "Smart, adaptative and autonomous sensing" (#497040) awarded to D. C. Côté. Stéphane Boudreau kindly performed the species identification of the shrubs used in this study.



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
