# Peer review of "Impact of shrub branches on the shortwave vertical irradiance profile in snow"

_EGUsphere, 2024_

## Author Comment (AC1)

*We thank the Reviewer for the time spent reading our paper and for giving us the opportunity to improve our text. In this review, we are often requested to repeat several times a number of statements. We believe that the concision requirements of scientific paper writing encourage writers to avoid repetitions. When repetitions are recommended by the reviewer, we therefore often choose not to follow such recommendations.*

*This paper is intended for readers of The Cryosphere, who are expected to be familiar with basic concepts of snow studies. Again for the sake of concision, recommendations by the reviewer to detail such concepts are not followed. Our responses are embedded in the Reviewer's comments, in blue italics.*

This study investigates the impact of shrub branches on irradiance by monitoring light levels at specific wavelengths (390±125 nm and >715 nm) within snow-covered shrub areas and adjacent grassland throughout a winter season. Light sensors were deployed at fixed heights within the shrubs and on the grassland. While the research presents interesting findings relevant to snow-related studies, certain areas require improvement to enhance the overall quality of the manuscript.

Major comments:

Q1: The manuscript's Abstract and Introduction currently lack a clear articulation of the research objectives and significance. It's crucial to explicitly state the research gaps addressed in this study compared to previous work. The reader should readily understand the motivation behind this research. Please revise and improve these sections accordingly.

*We apologize for this lack of clarity. Regarding the abstract, we indeed did not stress that no data was available to quantify the reduction of photochemical reaction rates caused by shrubs. We propose to add a sentence line 24 stating that "No study is currently available to quantify the reduction in photochemical rates caused by shrubs buried in snow. Here we quantify this effect by monitoring irradiance...".*

*Regarding the Introduction, we attempted to first make general statements on the impact of snowpack photochemistry, then explain that the role of impurities had been well studied, but that the role of shrubs had received little attention. We then logically proceed with the explanation of our project, intended to fill the gap. We propose to modify line 62 and explain that "To contribute to the understanding of shrub effects on irradiance profiles in snow, and to deduce the resulting impact on photochemical reaction rates in the snowpack and their potential consequences on*

*atmospheric chemistry, we have monitored….". We hope that these modifications will clarify our objectives for readers.*

Q2: While Section 3 presents numerous figures and tables, it lacks detailed descriptions and explanations for them. This makes it challenging for readers to understand the results. It's important to guide the readers through the findings and not leave them to guess the story behind the data. Please provide comprehensive descriptions and interpretations for all figures and tables.

*Section 3 contains Figures 5 to 11 and Tables 1 to 2. Figure 5 shows vertical profiles of snow physical properties, familiar to readers of The Cryosphere. Figure 6 shows time series of snow height which are also familiar to readers of The Cryosphere. It also shows the dates of field work and the dates for which simulations were performed, and lastly the height of sensors placed in snow which probably. All this is explained by a detailed and lengthy caption of 92 words. Figures 7 and 9 show time series of irradiance signals, explained by a caption of 85 words. In the caption we propose to replace "CNR4" with "CNR4 radiometer" to clarify the type of instrument. Figure 8 shows time series of irradiance signal during specific days. To clarify the use of these data, we propose to add that "For the February 28th data, the time range 10:30 to 12:30 was used. For the March 6th data, the time range 7:00 to 16:00 was used, with the exclusion of the 11:30-13:00 time range. Part of this is however already explained in the text lines 242-243. It may be better to add some text to avoid duplicating the text in Figure captions. Figures 10 and 11 show the simulations of irradiance profiles in snow. The caption box shows "TARTES" which are simulation profiles using the TARTES software, as explained in the text, but which may not be clear to readers just looking at Figures. To clarify this, we propose to rephrase the caption as follows "Profiles of irradiance in the snowpack at FIELD and SHRUB at 390 nm simulated by the TARTES software. Experimental data points are also shown. The scale…". Similar modifications to the caption of Figure 11 are proposed. Regarding Table 1, we propose modify the caption as follows "Canopy and branch characteristics at heights of 325, 485 and 650 mm heights, corresponding to the levels of the three sensors in SHRUB that were buried in snow". Regarding Table 2, we believe the caption adequately describes the Table, as all the variables listed are mentioned in the caption.*

Q3: The study simulates the influence of shrub branches using a "soot-equivalent" approach. However, figures like Fig. 2, Fig. 5, and Fig. S1-S3 highlight the variability of snowpack properties. Deep snow and high specific surface area (SSA) can significantly impact irradiance. The current analysis doesn't seem to account for these snowpack properties, leading to potentially inaccurate simulation results. This is particularly evident in Section 3.4, where the lack of

consideration for snowpack properties results in convoluted and confusing explanations. That means your conclusion in Abstract and Conclusion section would be modified. Please address this issue.

*We respectfully disagree with the reviewer. Table 2, which occupies a whole page (page 15) details our careful consideration of snowpack density and SSA profiles in simulations. The values used are based on our field measurements shown in great detail in Figures 5, S2 and S3. The variability on snowpack properties has therefore been at the core of our reasoning and simulations. Line 185, we explain regarding simulations that "In TARTES, input data are the thickness, density and SSA of each snow layer.", implying we do consider snowpack properties. Lines 265-266 we further state "An irradiance profile can be simulated if the physical properties (SSA and density) and impurity concentrations of the snow layers are known", again implying we do consider these properties. Subsequent lines stress even further than these properties are at the core of our simulations.*

Q4. Branch density is a crucial factor, yet it's only briefly mentioned in Section 4.3. I recommend including comparative tests in Section 3 to explore its influence.

*Branch density is indeed a crucial factor. As far as our work is concerned, branch density manifests by the amount of light absorbed. We explain in detail that branches are considered as a homogeneous absorber like soot, despite the fact that they are discrete absorbers. This is mentioned in the abstract line 28, in methods line 191, in the discussion lines 352, 397 and 404 and in the conclusion line 426. Therefore, in our approach variations in branch density will have the same effect as variation in soot concentrations. We propose to stress and clarify this by adding in the Methods section, line 193: "We therefore expect a higher branch density to manifest itself by requiring the use of a higher soot concentration in simulations".*

*The validity of this approach is challenged at 760 nm, and we therefore already present comparative tests in this case in section 3, as detailed in Figure 11 and lines 315-317: "We therefore performed simulations at 760 nm using the same soot concentrations for SHRUB 315 as at 390 nm. To investigate the impact of bark optical properties that would vary with wavelength differently to soot, we also performed simulations with concentration multiplied by 0.33 and by 3. Results of simulations at 760 nm are shown in Fig. 11"*

Minor comments:

Abstract

Lines 23-24: The sentence needs clarification and rephrasing.

The abstract should clearly highlight the research gap this study aims to fill.

*Indeed, we have made a suggestion in our reply to Q1.*

Introduction

Line 38-41: Please provide more information on this physical process.

*We propose to replace "Snowpack photochemistry modifies the snow composition and produces..." with "Chemical reactions in the snowpack lead to the production of numerous species which are released in snowpack interstitial air. Produced species include NO and $NO_2$ ...."*

Line 46-47: Explain the focus on the 300-450nm wavelength range. And comment on the use of 760 nm in this study.

*The focus on the 300-450 nm wavelength range is because "Most snowpack photochemical reactions are triggered by radiation in the 300 to 450 nm wavelength range (Grannas et al.,2007; Wang, 2021)", as explained lines 46-47. We also write in the abstract, lines 30-31: "Noting that photochemically active radiation is mostly in the near UV and blue...". Furthermore, we write line 64 "The 390 nm wavelength is within the most photoactive wavelength range...". Regarding the 760 nm wavelength range, we write lines 66-68: "At 760 nm, photochemistry is not known to be active for most molecules. However, at this wavelength, the ice absorption coefficient is about 120 times greater than at 390 nm (Picard et al., 2016), so that investigating this longer wavelength informs us on the impact of shrubs under more absorbing ice conditions."*

Line 54-55: Expand the introduction of previous studies, detailing their measurement methods and identifying research gaps they left unaddressed.

*Thank you for raising this point. In fact, these 3 studies measured the impact of shrubs protruding above the snow on irradiance above the snow. It is therefore not relevant to our study, focused on irradiance within the snowpack. Mentioning them adds confusion without any added value for our purpose. We will therefore delete the mention to these 3 studies. We will delete lines 54-58.*

Line 62: Specify the species of shrub studied.

*We will add Alnus incana, as already mentioned in the abstract (line 25) and in methods, line 100.*

Line 66: Add a reference to support your statement made.

*We will add the references (Grannas et al., 2007; Wang, 2021), already cited lines 46-47.*

2.2 Sensor deployment and site description

Figure 2: Include images to illustrate sensor deployment both before and after snow cover, showing how measurements are taken.

*Figure 2 shows such pictures before sensor head burial and after its burial. Figure S1 also provides an extra 4 pictures detailing the setup with views of the sensors before burial. After burial, sensors are not visible anymore, as shown in Figure 2.*

Section 2.4: change all instances of "snow heigh" to "snow depth"

*When the snow surface is used as a reference, snow depth is adequate. When the ground is used as a reference, snow height is more appropriate. Snow height is commonly used and we use it as required.*

Line 172: Provide an explanation for the statement made.

*We apologize for the lack of clarity. We propose to replace "For the SHRUB sensors, $(I_{r,i}/I_0)_c$ therefore takes into account the shading by shrub branches." With "When shrubs are present, branches protruding above the snow reduce the radiation incident on the snow surface, and this reduction appears in $(I_{r,i}/I_0)_c$. However, since we are interested in the extinction within the snowpack, this does not affect our data analysis."*

*In fact, we realize that this initially confusing statement adds no useful information. It is probably even better to simply delete it, which is our preferred option.*

Line 187: Specify the number of snow layers considered and describe how snow depth is divided into these layers.

*We could complement "In TARTES, input data are the thickness, density and SSA of each snow layer." With "In TARTES, input data are the thickness, density and SSA of each snow layer, as determined from observations." However, this appears very clearly when results are detailed, so this addition may not be useful at this stage.*

Lines 187-188 & 190-191: Justify the assumption that all absorbing impurities are soot-like and explain why other elements like dust are not considered.

*We do not assume that all absorbing impurities are soot or even soot-like but we seek a soot equivalent concentration in the range of wavelength of interest for photochemistry. This simplifies the problem "we consider for simplicity," (line 187) that is otherwise intractable without spectrometer measurements. Dust is certainly present, as well as numerous other absorbers, but all we need is an impurity absorption coefficient accounting for all absorbers. We could also treat it as dust, and come up with a dust equivalent. However, soot is often the main absorber in snow, and presents less diversity and has a flat absorption spectrum compared to any other components which make it more suitable for an equivalent. See for instance Tuzet et al. 2019.*

*Tuzet, F., Dumont, M., Arnaud, L., Voisin, D., Lamare, M., Larue, F., Revuelto, J., and Picard, G.: Influence of light-absorbing particles on snow spectral irradiance profiles,* The Cryosphere, *3, 2169–2187, doi:10.5194/tc-13-2169-2019, 2019*

*To clarify this, we propose to add line 193: "Any other type of impurity could be used to simulate absorption. The important parameters are the optical constant and the concentration of the impurity, which determine absorption."*

Line 195 and 202: Explain how the values "~29 cm" and "8.2 cm" were derived.

*We propose to replace lines 193-194: "At 390 nm, we calculate that for typical snow encountered during this study (density=200 kg m-3, SSA=25 m2 kg-1, soot=25 ng g-1), irradiance is reduced by a factor of 10 over a distance of 29 cm." with "At 390 nm, we calculate using TARTES that irradiance is reduced by a factor of 10 at a depth of 29 cm for typical snow encountered during this study (density=200 kg $m^{-3}$, SSA=25 $m^2$ $kg^{-1}$, soot=25 ng $g^{-1}$).*

Section 3:

Line 214-215 & 221-222: Provide more detailed explanations for Fig. 5 and Fig. 6, guiding the reader through the snowpack properties evolution and the significance of the figures.

*These Figures are intended for readers of The Cryosphere, who have at least minimal familiarity with snow studies. Figure 5 shows vertical profiles of snow physical properties, among the most basic data shown in snow field studies. Likewise, Figure 6 is a time series of snow height, again a basic concept in snow field studies.*

Line 232-234 & 244: Clarify the conclusions drawn and specify the variables or evidence supporting them.

*This paper was written with the understanding that readers had minimal knowledge regarding a time series of solar irradiance during daytime under clear-sky conditions. We explain, referring to Figure 7, that the data coming from the monitoring of downwelling solar radiation on March 3[rd] is characteristic of clear-sky conditions, which we feel will be obvious to readers of The Cryosphere. On other days, plots differ from this shape. Irregular variations in irradiance indicate variable cloudiness while days with permanent low irradiance indicate continuous cloudiness. We expect this to be known by readers.  Line 233 we could add "solar" before "shortwave irradiance" and "radiometer" after "CNR4" if that would help. Regarding line 244, we feel that the current text "As expected, irradiance signals are lower at SHRUB than at FIELD because of light absorption by shrub branches" will be understood by readers because we already discuss this at length in the Introduction and in Methods.*

Line 236: Address the potential uncertainty error in the simulation due to direct radiation on March 6[th]

*Line 236, we will add "Periods with direct radiations were removed from the analysis."*

Line 245: Explain the selection of specific days for analysis and clarify the statements made in relation to Section 3.1.

*Line 177, we state clearly that we "limit our data analysis to overcast conditions, when incident light was diffuse, similar to the conditions of the sensors buried in the snow.". Therefore, the days selected were overcast. We will nevertheless add line 245 "because overcast conditions were observed".*

Line 245: why did you select these four days "February 2nd, 3rd and 23rd and April 1st" for analysis? If you think the following sentence is the reason, it is still unclear. You didn't give the explanation in Section 3.1

*The explanation was given line 177 and as just stated, we will repeat it here.*

Line 245-246: Explain the statement made here.

*We understand the reviewer is referring to the statement ". Only one to two sensors were then buried, as visible in Fig. 6." We believe that a cursory look at Figure 6, which shows the time series of snow height and the height of the sensors, will*

*convince the attentive reader that on the days discussed, indeed one to 2 sensors were buried.*

Line 253: Provide additional explanation and comment for Fig. 9.

*This is similar to Figure 7, but for the red radiation. We explain line 67 that ice absorbs much more at red wavelengths than at blue wavelengths, and we repeat this line 251. Lower signals are expected, especially at depth. We believe this is understandable for readers of The Cryosphere.*

Section 3.3: "A... reported in Table 1. B.... is shown in Fig. S4. ...". Clarify the purpose of the two sentences and Table 1 in this section.

*We will change "The mean diameter and number of branches of the shrub canopy in a representative shrub at the level of the S325, S485 and S650 sensors are reported in Table 1." To "The mean diameter and number of branches of the shrub canopy in a representative shrub at heights of 325, 485, and 650 mm, which correspond to the heights of the S325, S485 and S650 sensors are reported in Table 1." We will change "The distribution of branch diameters at these same levels is shown in Fig. S4" to "The distributions of branch diameters at these three heights are shown in Fig. S4".*

Line 266: Explain the selection of these specific days for analysis "February 2nd, 3rd, 23rd and 28th, March 6th and April 1st."

*These were overcast days, as explained twice in the text above.*

Line 268-271: Rephrase the sentence to improve clarity on the simulation parameters used.

*We propose to change "For February $2^{nd}$ and $23^{rd}$, we used the physical data obtained on those very days during our snowpit measurements." With "For February $2^{nd}$ and $23^{rd}$, we used the snow density and specific surface area values obtained on those very days during our snowpit measurements."*

Line 272-273: Provide references or evidence to support the idea presented.

*Our text reads: "The concentration of impurities in the snow, treated as soot-equivalent, was not measured and was used as an adjustable variable". This is a methodological choice, as explained in the methods section, lines 187-188, which reads "Here, we consider for simplicity that all absorbing impurities are soot, with properties reported in Bond and Bergstrom, (2006),"*

Table 2: Clarify if the soot density information is derived from the simulation, based on the description in Lines 272-273.

*Yes, as the Reviewer mentions, this has already been described lines 272-273 and we will not repeat it here.*

Fig. 10: Provide further descriptions and comments to guide the reader's understanding.

*We addressed this comment in the Reviewer's Q2 comment and will not repeat this here.*

Line 356: "Figs. 9 and 10 illustrate that irradiance decreases faster with depth at SHRUB than at FIELD." Acknowledge that the faster decrease in irradiance with depth at SHRUB compared to FIELD also suggests the influence of snow properties on irradiance reduction

*We are not sure to understand this comment. Perhaps the Reviewer is suggesting that irradiance reduction is also caused by snow, as a function of its density and specific surface area. This basic snow physics concept has been alluded to many times in the text and need not be repeated here. Furthermore, we are only discussing the comparison between SHRUB and FIELD, so we do not feel this comment is relevant to this part of the discussion.*

---

## Author Response (AR1)

**Responses to Review 1**

*We thank the Reviewer for the time spent reading our paper and for giving us the opportunity to improve our text. In this review, we are often requested to repeat several times a number of statements. We believe that the concision requirements of scientific paper writing encourage writers to avoid repetitions. When repetitions are recommended by the reviewer, we therefore often choose not to follow such recommendations. Our responses are embedded in the Reviewer's comments, in blue italics. Line numbers refer to those in the tracked changes version.*

This study investigates the impact of shrub branches on irradiance by monitoring light levels at specific wavelengths (390±125 nm and >715 nm) within snow-covered shrub areas and adjacent grassland throughout a winter season. Light sensors were deployed at fixed heights within the shrubs and on the grassland. While the research presents interesting findings relevant to snow-related studies, certain areas require improvement to enhance the overall quality of the manuscript.

Major comments:

Q1: The manuscript's Abstract and Introduction currently lack a clear articulation of the research objectives and significance. It's crucial to explicitly state the research gaps addressed in this study compared to previous work. The reader should readily understand the motivation behind this research. Please revise and improve these sections accordingly.

*We apologize for this lack of clarity. Regarding the abstract, we indeed did not stress that no data was available to quantify the reduction of photochemical reaction rates caused by shrubs. We added a sentence line 24 stating that "No study is currently available to quantify the reduction in photochemical rates caused by shrubs buried in snow. Here we quantify this effect by monitoring irradiance...".*

*Regarding the Introduction, we attempted to first make general statements on the impact of snowpack photochemistry, then explain that the role of impurities had been well studied, but that the role of shrubs had received little attention. We then logically proceed with the explanation of our project, intended to fill the gap. We modified line 65 and explain that "To contribute to the understanding of shrub effects on irradiance profiles in snow, and to deduce the resulting impact on photochemical reaction rates in the snowpack and their potential consequences on atmospheric chemistry, we have monitored....". We hope that these modifications will clarify our objectives for readers.*

Q2: While Section 3 presents numerous figures and tables, it lacks detailed descriptions and explanations for them. This makes it challenging for readers to understand the results. It's important to guide the readers through the findings and not leave them to guess the story

behind the data. Please provide comprehensive descriptions and interpretations for all figures and tables.

*Section 3 contains Figures 5 to 11 and Tables 1 to 2. Figure 5 shows vertical profiles of snow physical properties, which are basic snow data familiar to readers of The Cryosphere. Figure 6 shows time series of snow height which are also familiar to readers of The Cryosphere. The Figure is explained by a detailed and lengthy caption of 92 words. Figures 7 and 9 show time series of irradiance signals, explained by a caption of 85 words. In the caption we replaced "CNR4" with "CNR4 radiometer" to clarify the type of instrument. Figure 8 shows time series of irradiance signal during specific days. To clarify the use of these data, we added that "For the February 28th data, the time range 10:30 to 12:30 was used. For the March 6th data, the time range 7:00 to 16:00 was used, with the exclusion of the 11:30-13:00 time range." In the caption. Please note that this is already explained in the text lines 260-262, so we are unsure this addition is necessary. Figures 10 and 11 show the simulations of irradiance profiles in snow. The caption box shows "TARTES" which are simulation profiles using the TARTES software, as explained in the text, but which may not be clear to readers just looking at Figures. To clarify this, we rephrased the caption as follows "Profiles of irradiance in the snowpack at FIELD and SHRUB at 390 nm simulated by the TARTES software. Experimental data points are also shown. The scale...". Similar modifications to the caption of Figure 11 were done. Regarding Table 1, we modified the caption as follows "Canopy and branch characteristics at heights of 325, 485 and 650 mm heights, corresponding to the levels of the three sensors in SHRUB that were buried in snow". Regarding Table 2, we believe the caption adequately describes the Table, as all the variables listed are mentioned in the caption.*

Q3: The study simulates the influence of shrub branches using a "soot-equivalent" approach. However, figures like Fig. 2, Fig. 5, and Fig. S1-S3 highlight the variability of snowpack properties. Deep snow and high specific surface area (SSA) can significantly impact irradiance. The current analysis doesn't seem to account for these snowpack properties, leading to potentially inaccurate simulation results. This is particularly evident in Section 3.4, where the lack of consideration for snowpack properties results in convoluted and confusing explanations. That means your conclusion in Abstract and Conclusion section would be modified. Please address this issue.

*We respectfully disagree with the reviewer. Table 2, which occupies a whole page (page 15) details our careful consideration of snowpack density and SSA profiles in simulations. The values used are based on our field measurements shown in great detail in Figures 5, S2 and S3. The variability on snowpack properties has therefore been at the core of our reasoning and simulations. Line 185, we explain regarding simulations that "In TARTES, input data are the thickness, density and SSA of each snow layer.", implying we do consider snowpack properties. Lines 286-287 we further state "An irradiance profile can be simulated if the physical properties (SSA and density) and impurity concentrations of the snow layers are known", again implying we do consider these properties. Subsequent lines stress even further than these properties are at the core of our simulations.*

Q4. Branch density is a crucial factor, yet it's only briefly mentioned in Section 4.3. I recommend including comparative tests in Section 3 to explore its influence.

*Branch density is indeed a crucial factor which manifests by the amount of light absorbed. We explain in detail that we make the hypothesis that branches are considered as a homogeneous absorber like soot, despite the fact that they are discrete absorbers. This is mentioned, necessarily briefly, in the abstract line 29. In Methods, we now elaborate on this and clarify this point. Lines 197-201, we now write "What TARTES uses is an absorption coefficient, which can be translated into a concentration of any impurity provided that the mass absorption coefficient of that impurity is known. It may also eventually be translated into a branch density.  Here, we translate for simplicity the absorption coefficient used in TARTES into a soot concentration, because soot is a common and highly absorbing impurity in snow (Hansen and Nazarenko, 2004; Chylek et al., 1983; Warren and Clarke, 1990). The soot optical properties used are those reported in (Bond and Bergstrom, 2006)". This, combined with the following lines 203-203, should make it clear that we consider branch density a crucial factor.*

*Branch density is also mentioned in the discussion lines 383, 418-423 and in the conclusion line 446. The requested comparative test was already made in Table 4.*

Minor comments:

Abstract

Lines 23-24: The sentence needs clarification and rephrasing.

The abstract should clearly highlight the research gap this study aims to fill.

*This repeats Q1, which has been addressed above.*

Introduction

Line 38-41: Please provide more information on this physical process.

*We replaced "Snowpack photochemistry modifies the snow composition and produces..." with "Chemical reactions in the snowpack lead to the production of numerous species which are released in snowpack interstitial air. Produced species include NO and $NO_2$ ....", lines 39-41*

Line 46-47: Explain the focus on the 300-450nm wavelength range. And comment on the use of 760 nm in this study.

*The focus on the 300-450 nm wavelength range is because "Most snowpack photochemical reactions are triggered by radiation in the 300 to 450 nm wavelength range (Grannas et al.,2007; Wang, 2021)", as explained lines 49-50. We also write in the abstract, lines 31-32: "Noting that photochemically active radiation is mostly in the near UV and blue...". Furthermore, we write line 68 "The 390 nm wavelength is within the most photoactive wavelength range...". Regarding the 760 nm wavelength range, we write lines 66-68: "At 760 nm, photochemistry is not known to be active for most molecules. However, at this wavelength, the ice absorption coefficient is about 120 times greater than at 390 nm (Picard et al., 2016), so that investigating this longer wavelength informs us on the impact of shrubs under more absorbing ice conditions." We believe these explanations are sufficient.*

Line 54-55: Expand the introduction of previous studies, detailing their measurement methods and identifying research gaps they left unaddressed.

*Thank you for raising this point. In fact, these 3 studies measured the impact of shrubs protruding above the snow on irradiance above the snow. It is therefore not relevant to our study, focused on irradiance within the snowpack. Mentioning them adds confusion without any added value for our purpose. We will therefore delete the mention to these 3 studies. Lines 56-61 have been deleted.*

Line 62: Specify the species of shrub studied.

*We added Alnus incana line 67, as already mentioned in the abstract (line 26) and in methods, line 105.*

Line 66: Add a reference to support your statement made.

*We added the references (Grannas et al., 2007; Wang, 2021).*

2.2 Sensor deployment and site description

Figure 2: Include images to illustrate sensor deployment both before and after snow cover, showing how measurements are taken.

*Figure 2 shows such pictures before sensor heads burial and after its burial. Figure S1 also provides an extra 4 pictures detailing the setup with views of the sensors before burial. After burial, sensors are not visible anymore, as shown in Figure 2. We believe these images are sufficient.*

Section 2.4: change all instances of "snow heigh" to "snow depth"

*When the snow surface is used as a reference, snow depth is adequate. When the ground is used as a reference, snow height is more appropriate. Snow height is commonly used and we use it as required.*

Line 172: Provide an explanation for the statement made.

*We realize that this statement can be confusing. Furthermore, it adds no useful information. We deleted it, line 180.*

Line 187: Specify the number of snow layers considered and describe how snow depth is divided into these layers.

*We complemented "In TARTES, input data are the thickness, density and SSA of each snow layer." With "In TARTES, input data are the thickness, density and SSA of each snow layer, as determined from observations.", line 197. This also appears very clearly when results are detailed, as well as the division into snow layers.*

Lines 187-188 & 190-191: Justify the assumption that all absorbing impurities are soot-like and explain why other elements like dust are not considered.

*This comment is similar to Q4 above. We do not assume that all absorbing impurities are soot or even soot-like but we seek a soot equivalent concentration in the range of wavelength of interest here. To clarify this, we have added lines 197-201 "What TARTES uses is an absorption coefficient, which can be translated into a concentration of any impurity provided that the mass absorption coefficient of that impurity is known. It may also eventually be translated into a branch density. Here, we translate for simplicity them absorption coefficient used in TARTES into a soot concentration, because soot is a common and highly absorbing impurity in snow (Hansen and Nazarenko, 2004; Chylek et al., 1983; Warren and Clarke, 1990). The soot optical properties used are those reported in (Bond and Bergstrom, 2006)"*

Line 195 and 202: Explain how the values "~29 cm" and "8.2 cm" were derived.

*We replaced lines 194-195: "At 390 nm, we calculate that for typical snow encountered during this study (density=200 kg m-3, SSA=25 m2 kg-1, soot=25 ng g-1), irradiance is reduced by a factor of 10 over a distance of 29 cm." with "At 390 nm, we calculate using TARTES that irradiance is reduced by a factor of 10 at a depth of 29 cm for typical snow encountered during this study (density=200 kg $m^{-3}$, SSA=25 $m^2$ $kg^{-1}$, soot=25 ng $g^{-1}$). Lines 207-209.*

Section 3:

Line 214-215 & 221-222: Provide more detailed explanations for Fig. 5 and Fig. 6, guiding the reader through the snowpack properties evolution and the significance of the figures.

*Figure 5 just shows vertical profiles of snow physical properties. These are common plots seen in numerous papers appearing in The Cryosphere. Perhaps the reviewer is suggesting to comment variations, metamorphism, etc. This will not serve our purpose and again, for concision, we will just*

*focus on our topic: the role of shrubs on irradiance profiles. Likewise, Figure 6 is just time series of snow height, very common in papers in The Cryosphere. There is again no need to detail the precipitation and melting events at this stage. We do comment melting events when required, e.g., section 4.1. However, here, extra comments are not useful to our purpose and we do want to remain as concise as possible.*

Line 232-234 & 244: Clarify the conclusions drawn and specify the variables or evidence supporting them.

*There are no conclusions here, just a factual description of the data. We explain, referring to Figure 7, that the data coming from the monitoring of downwelling solar radiation on March 3$^{rd}$ is characteristic of clear-sky conditions, which we feel is pretty clear. On other days, plots differ from this shape. Irregular variations in irradiance indicate variable cloudiness while days with permanent low irradiance indicate continuous cloudiness. We feel these are simple facts. To make sure there is no confusion, line 249 we added "solar" before "shortwave irradiance" and "radiometer" after "CNR4". Regarding line 244 (now 261), we feel that the current text "As expected, irradiance signals are lower at SHRUB than at FIELD because of light absorption by shrub branches" will be understood by readers because we already discuss this at length in the Introduction and in Methods.*

Line 236: Address the potential uncertainty error in the simulation due to direct radiation on March 6$^{th}$

*Line 252, we added "Periods with direct radiations were removed from the analysis."*

Line 245: Explain the selection of specific days for analysis and clarify the statements made in relation to Section 3.1.

*Line 186, we state clearly that we "limit our data analysis to overcast conditions, when incident light was diffuse, similar to the conditions of the sensors buried in the snow.". Therefore, the days selected were overcast. We nevertheless repeated line 263 "because overcast conditions were observed".*

Line 245: why did you select these four days "February 2nd, 3rd and 23rd and April 1st" for analysis? If you think the following sentence is the reason, it is still unclear. You didn't give the explanation in Section 3.1

*The explanation was given line 185-187 "it was simpler and probably less error-prone to limit our data analysis to overcast conditions, when incident light was diffuse, similar to the conditions of the sensors buried in the snow" and as just stated, has been repeated line 263.*

Line 245-246: Explain the statement made here.

*We understand the reviewer is referring to the statement ". Only one to two sensors were then buried, as visible in Fig. 6." We believe that a cursory look at Figure 6, which shows the time series of snow height and the height of the sensors, will convince the attentive reader that on the days discussed, indeed one to 2 sensors were buried.*

Line 253: Provide additional explanation and comment for Fig. 9.

*This is similar to Figure 7, but for the red radiation. We explain line 71 that ice absorbs much more at red wavelengths than at blue wavelengths, and we repeat this line 269. Lower signals are expected, especially at depth.*

Section 3.3: "A... reported in Table 1. B.... is shown in Fig. S4. ...". Clarify the purpose of the two sentences and Table 1 in this section.

*We changed "The mean diameter and number of branches of the shrub canopy in a representative shrub at the level of the S325, S485 and S650 sensors are reported in Table 1." To "The mean diameter and number of branches of the shrub canopy in a representative shrub at heights of 325, 485, and 650 mm, which correspond to the heights of the S325, S485 and S650 sensors are reported in Table 1." (line 277). We changed "The distribution of branch diameters at these same levels is shown in Fig. S4" to "The distributions of branch diameters at these three heights are shown in Fig. S4". (line 279).*

Line 266: Explain the selection of these specific days for analysis "February 2nd, 3rd, 23rd and 28th, March 6th and April 1st."

*These were overcast days, as explained twice in the text above.*

Line 268-271: Rephrase the sentence to improve clarity on the simulation parameters used.

*We replaced "For February $2^{nd}$ and $23^{rd}$, we used the physical data obtained on those very days during our snowpit measurements." with "For February $2^{nd}$ and $23^{rd}$, we used the snow density and specific surface area values obtained on those very days during our snowpit measurements (Figure 5 and Figure S2)." (Line 288).*

Line 272-273: Provide references or evidence to support the idea presented.

*Our text reads: "The concentration of impurities in the snow, treated as soot-equivalent, was not measured and was used as an adjustable variable". This is a methodological choice, as now explained in great detail in the methods section, lines 197-202, as discussed earlier.*

Table 2: Clarify if the soot density information is derived from the simulation, based on the description in Lines 272-273.

*Yes, as the Reviewer mentions, this has already been clearly mentioned lines 292-293. "The concentration of impurities in the snow, treated as soot-equivalent, was not measured and was used as an adjustable variable". and we will not repeat it here. A Table heading has to be kept short and a Table has to be read with the corresponding text.*

Fig. 10: Provide further descriptions and comments to guide the reader's understanding.

*We addressed this comment in the Reviewer's Q2 comment and will not repeat this here.*

Line 356: "Figs. 9 and 10 illustrate that irradiance decreases faster with depth at SHRUB than at FIELD." Acknowledge that the faster decrease in irradiance with depth at SHRUB compared to FIELD also suggests the influence of snow properties on irradiance reduction

*We are not sure to understand this comment. Perhaps the Reviewer is suggesting that irradiance reduction is also caused by snow, as a function of its density and specific surface area. This basic snow physics concept has been alluded to many times in the text and need not be repeated here. Furthermore, we are only discussing the comparison between SHRUB and FIELD, so we do not feel this comment is relevant to this part of the discussion.*

**Responses to Review 2**

*We thank the Reviewer for the time spent reading our paper and for providing useful comments. Our responses are embedded in the Reviewer's comments, in blue italics. Line numbers refer to those in the tracked changes version.*

Like soot, shrub branches can have large impacts on the vertical irradiance distribution in the snowpack. This study used the comparative measurements of irradiance in snow with and without shrub branches to show the significant impacts of shrub branches in radiative transfer processes in snowpack, which is interesting and promising. My most comments are related to the technical clarifications and in-depth discussion. Please see below for my specific comments.

Major concerns:

1. Section 2.4: Please provide more details on the snow height estimation. How did the authors use camera photos to estimate snow height? Please also provide more details on the uncertainty of 1.5 cm.

*We changed "The snow height was determined from camera photos taken on site" to "An image analysis software which determined the snow level on the striped poles was used to determine snow height from photographs taken by a time-lapse camera". Lines 159-160. Regarding the 1.5 cm uncertainty, we changed "Considering spatial variations, we estimate the uncertainty on snow height to be 1.5 cm." to "Spatial variations between the various striped poles indicate an uncertainty on snow height at the sensors of interest of 1.5 cm." Lines 163-164.*

2. Line 170-174: How did the authors acquire the gain correction factor as well as the detection limit?

*Line 178, we changed "The analysis of data when no sensor was buried" to "The comparison of signals from all sensors when no sensor was buried". Regarding the detection limit, we changed "The detection limit for $I_{r,i}/I_0$ was found to be 0.002" to "The detection limit for $I_{r,i}/I_0$ was taken as the signal greater than 3 times the noise, and found to be 0.002." Line 180.*

3. Line 175-194: How did the authors determine the sky conditions as the overcast?

*Line 191-192, we changed "Overcast conditions were determined from time lapse images and from irradiance values" to "Overcast conditions were determined from time lapse images, which revealed the presence of clouds and the lack of shadows, and from the daily time series of solar irradiance, which deviated from the typical clear-sky plots". Subsequently, Figure 7 illustrates the various sky conditions.*

4. Line 187-190: Please clarify how the authors considered the soot in the simulations, considering no direct measurements of soot concentration? How about dust? Black carbon, brown carbon and dust can have very distinct impacts on snowpack.

*We now explain lines 197-202 that all we need in the simulations is an absorption coefficient regardless of the type of impurities: "What TARTES uses is an absorption coefficient, which can be translated into a concentration of any impurity provided that the mass absorption coefficient of that impurity is known. It may also eventually be translated into a branch density. Here, we translate for simplicity the absorption coefficient used in TARTES into a soot concentration, because soot is a common and highly absorbing impurity in snow (Hansen and Nazarenko, 2004; Chylek et al., 1983; Warren and Clarke, 1990). The soot optical properties used are those reported in (Bond and Bergstrom, 2006)".*

*Subsequently, lines 292-293 explain "The concentration of impurities in the snow, treated as soot-equivalent, was not measured and was used as an adjustable variable to optimize the agreement between measured and simulated irradiance profiles."*

5. Line 190: Please clarify why did the authors can use soot to simulate the impacts of branches? Their optical properties can be very different.

*We believe our addition lines 197-202 also addresses adequately this question.*

6. Line 196-208: There are many numbers in this paragraph. Please clarify where these numbers are from and what is the major points from these numbers?

*To clarify our objective, we now start this paragraph with "Figure 4 sums up the optical properties of the constituents considered here: ice, soot and bark". Line 211.*

7. Section 3.1 & 3.2, 4.2: Please add the statistical tests to check whether the differences are significant among SHRUB and FIELD. Besides, the introduction in section 3.1 is too simple.

*The objective of this work is to determine the soot equivalent of shrub branches. Table 2 shows rather clearly that the soot values of SHRUB are always greater than those of FIELD. Statistics do not appear as a necessary addition. For each line the soot value at SHRUB is greater than that of FIELD. The average soot value of SHRUB is 111±54 ppb while at FIELD it is 23± 9 ppb. Discussing this further does not seem essential.*

*In section 3.1, we just factually report snow physical measurements, the methods of which have been detailed in section 2.5. The methods are standard, very common and well established in snow field studies. We do not think extra details are required here.*

8. Section 3.3 is too simple. Did the authors use these data for the analysis and explanations?

*Yes indeed, we do want to keep this simple and concise. The only objective of this section is to simply present data on branch numbers and diameters. This must remain simple. These data are used in the discussion, section 4.4.*

9. Line 278-280: Please explain why the soot concentration of a given layer is not expected to vary significantly over time.

*We added line 299 "because soot is not significantly affected by snow metamorphism and by snow chemistry, and soot particles are hydrophobic and little affected by melting events (Festi et al., 2021; Meyer and Wania, 2011)."*

10. Section 4.1: Can the authors provide some field-based evidences for this?

*The only true field evidence would have been to dig a snowpit at the very spots involved, but this destructive action is not compatible with our monitoring activity. Moreover, it would have required analyzing the data before the data were obtained. We do however provide field evidence at a nearby site. Fig. 6 of Bouchard et al. (2024) is a clear demonstration of the reality of the process invoked, as mentioned line 350. Percolation channels are very common in temperate snowpacks and are abundantly described in the literature (e.g. Sturm et al, 1995, https://doi.org/10.1175/1520-0442(1995)008<1261:ASSCCS>2.0.CO;2).*

11. Many parts of the discussion should be moved to results section.

*Indeed, this is debatable. Our choice however was to have all these parts in the discussion, because they are not derived from primary data analysis, but secondary data analysis. There are several possible strategies to write a paper.*

12. How the site-scale findings in this study can be extended and incorporated into Earth system models deserves more discussion.

*We feel that section 4.5 "Impact of branches on snowpack photochemistry", is already quite lengthy and we already mention line 424 that "We therefore speculate...", indicating that we already integrate a certain degree of speculation. We conclude this section by writing line 442 "The reduction of snow photochemical rates by shrub expansion may thus lead to numerous chemical and climatic effects that may deserve further quantification." We added at the end "using coupled models of snow and atmospheric chemistry. (Toyota et al., 2014; Zatko et al., 2016)." Going beyond that and discussing Earth system model would be overspeculation.*

Minor concerns:

1. Figure 3: Please provide the full names of some abbreviations in the caption.

*We added: PWR; Power; SSH: Secure Shell protocol; RSSH:  Reverse SSH protocol; SFTP: SSH File Transfer Protocol.*

2. Figure 4: How about the bark absorption?

*The data available in the literature is the bark reflectance. With the assumption that the bark is not transparent i.e., thick enough: 1-reflectance is equal to absorption. We changed the legend to 'bark absorptivity" and added in the caption: "The absorptivity of Alnus incana bark is calculated as 1 – bark reflectivity (Juola et al., 2022b), assuming bark is thick enough to be fully opaque".*

3. Table 2 & 4: How did the authors determine these values?

*Table 2: these values are discussed at length in the text. Lines 287-294 explain: "For February 2nd and 23rd, we used the snow density and specific surface area values obtained on those very days during our snowpit measurements (Figure 5 and Figure S2). For the other days, snow physical properties were estimated from the snowpit data, the literature (Domine et al., 2007; Taillandier et al., 2007) which helps in estimating the time-evolution of snow physical properties and the SSA-density correlation, and above all from our experience of snow physical properties and their evolution at the Montmorency Forest (Bouchard et al., 2023; Bouchard et al., 2022). The concentration of impurities in the snow, treated as soot-equivalent, was not measured and was used as an adjustable variable to optimize the agreement between measured and simulated irradiance profiles." This text appears just before Table 2.*

*Table 4: Lines 393-395 read: "Based on photographs of the S325 and S485 sensors taken during installation (Fig. S1) and also 370 on the data of Fig. S4 in the Supplement, we attempt to estimate the number and mean diameter of branches within two e-folding depths of each sensor, for both wavelengths studied. These estimates are reported in Table 4." This text appears just before Table 4.*

4. Section 4.4: This section belongs to results.

*We have addressed this comment in our response to major comment 11 above.*

---

## Author Response (AR2)

**Responses to reviews by the Editor and Reviewers**

*Thank you for these comments and recommendations. Our responses are in blue italics.*

**Editor's review**

Please address comments from the second round of reviews. I'll note that both reviewers ranked the scientific rigour and presentation quality of the paper as either fair or poor - indicating room for improvement in the manuscript that I encourage the authors to seriously consider.

The red sensor needs better justification of inclusion in the abstract and introduction, because this is really a paper about irradiance profiles in snow, and effective absorption by branches, but the presented motivation is photochemical reactions - which do not occur in the wavelengths for the 'red' sensor included in this study.

*Indeed, red radiation is not relevant to photochemistry. We had briefly mentioned metamorphism as a possible application for irradiance profiles (line 43). While our primary focus remains on photochemistry, we have developed the metamorphism aspect a bit more. Studying red radiation is of potential interest for metamorphism, as explained in the detailed responses to your comments.*

The Abstract

You focus it around photochemical reaction but, again, it is known that this isn't relevant for the longer 'red' sensor - so clearly state what motivated included a red sensor. Move up the description that photochemical reactions are triggered by solar radiation in the UV and blue wavelengths, but also state that ice is more absorptive in the red and makes for an interesting comparison. Use this to set up the chosen study wavelengths, and be descriptive, i.e. "Here we monitored irradiance at blue (390±125 nm) and red (715 - 1000 nm, effective 760 nm) wavelengths..." And then you can use red and blue in the abstract for clarity rather then different wavelength descriptors for red: "In the blue wavelengths, dense shrub branches were found to reduce irradiance similarly to about 140 ppb of soot. For the red wavelength, insufficient data and the greater ice absorption do not allow accurate conclusions."

*Thank you for these suggestions. We now mention metamorphism as another possible application in the abstract, for which spectral information is relevant. When mentioning photochemistry, we only refer to the 390 nm data. The 760 nm data is useful for quantifying energy absorption, which is relevant to metamorphism. We have restructured the abstract around both these topics so that the use of the 760 nm wavelength hopefully appears useful.*

You also need to set up inclusion of the red sensor in the introduction better, it's not until the 3rd paragraph that mention red wavelengths are not relevant for photochemistry - but this paper is about measure blue and red irradiance profiles - so... why?

*As mentioned in the abstract, irradiance profiles affect the radiative energy absorption in the snowpack, and hence metamorphism. We now briefly develop this, lines 52-54. We then discuss that radiation absorption in snow is highly wavelength-dependent and that therefore, for metamorphism-related applications, the spectral dependence of shrubs impact must also be studied (lines 73-77).*

Please report sensor heights in cm (to align Figure 6) - at the scale of tens of cm, using cm is more suitable

*This has been changed throughout. mm are not used anymore for sensor heights.*

Figure 6 could be improved for clarity and interpretation - it is busy and using red/orange and green as primary colors on a plot makes challenging for those with the most common form of color blindness to interpret the plot, please update. Don't indicate fieldwork timing that was not included in the study, it is irrelevant. Consider moving the sensor depths on the secondary axis and not overlaid on plot.

*Indeed, Figure 6 was busy. We have split it into 2 panels for clarity and changed the colors so that it is color blind-friendly. We have also diversified symbols to facilitate curve identification. The caption has been modified accordingly. We have removed all mentions of the January 6th field work, since the data were not used.*

Results

Please include an overview paragraph at the start of the Section Results summarizing the relevant findings and outline how the results will be presented to help guide readers about the results sections. For example, right now it's not clear why snow properties from February 23rd are presented first - there is no context.

*We have added an introductory paragraph to explain to the reader which data are presented and the general strategy, lines 233-238. This should make it easier for the reader to follow the structure of the data presentation. We have also made additional modifications later in the text to specify context. For example, line 240, we mention than snow field measurements were performed on three dates, but that we are only showing data from one date in the main text. We have also added context to the caption of Figure 5.*

Section 3.2 is not a meaningful results presentation - its primarily justifying conditions and discussing methods. This is basically the focus of the measurement methods - don't assume readers will interpret plots as you expect - what you would you like readers to take away? Please summarize and state results clearly.

*Section 3.2 is really about showing irradiance data, as evidenced by the presentation of Figures 7, 8 and 9, which all show such data. There is indeed some justification in the text, but this is in fact explanations of the figures and of the data selection process. This text requires the prior presentation of these data, and therefore these aspects could not be developed in the Methods section. What could be mentioned in Methods was in fact mentioned there, such as the choice of overcast days for simulations. Here we show the actual strategy with data. In fact this was partly in response to the reviewers' earlier request for explanations. Reviewer 1 did not seem to understand how the CNR4 data were used to select overcast days. In any case, we have added a paragraph for more context and clarity, lines 261-267. We have also modified all the Figures of this section to make them color blind-friendly. Moreover, we have modified Figure 9 to show that at 760 nm light transmission is much shallower. The message is that radiation attenuation is much greater in shrubs (Fig. 7a vs 7b and 8ab vs 8cd) and that attenuation is much greater at 760 than at 390 nm (Fig 7 vs Fig 9). This is stressed line 283 "As expected, irradiance signals are lower at SHRUB than at FIELD because of light absorption by shrub branches.", and lines 293-294 "Because of the greater ice absorption at 760 nm than at 390 nm (Warren and Brandt, 2008), radiation penetration is much shallower at 760 than at 390 nm and a signal was detected only for the topmost sensor at the red wavelength.". The actual take-home messages of the paper are in fact in the discussion, the main one being that photochemical rates are divided by 2 in shrubs.*

You didn't measure black carbon in snow but it's persistent and variable through both space and time - how would that impacts the analysis?

*Indeed, soot is persistent because it is not chemically reactive and it varies with snow layers because of variations in air mass composition. The snow composition reflects that of the air mass where it formed. However, in a given snow layer, soot is not expected to vary over time unless intense melting occurs. We stress this, lines 322-324 "However, the soot concentration of a given layer is not expected to vary significantly over time because soot is not significantly affected by snow metamorphism and by snow chemistry, and soot particles are hydrophobic and little affected by melting events (Festi et al., 2021; Meyer and Wania, 2011)." Our analysis respects this by adjusting soot concentrations for each layer, and then maintaining those concentrations constant over time. For the April $1^{st}$ simulations, maintaining soot values constant yields unsatisfactory simulations, as mentioned lines 330-331 "Using lower soot values on that last date [April $1^{st}$] would allow a perfect fit, but decreasing soot values during melt would not make physical sense.", and visible in Fig. 10, bottom right panel. We mention line 330 that "We reflect on this situation in the discussion.". Subsequently, in the discussion, section 4.1 line 370 explains that this is probably due to the formation of percolation channels, so that simulations using a plane-parallel geometry cannot reproduce the data.*

*Lastly, since we stress metamorphism as an extra field of application of our data in the abstract and introduction, we also added section 4.6 in the discussion to review the implications of our results for this subject. We also briefly mention metamorphism in the conclusion.*

**Report 1 by Reviewer 2**

Thank the authors to resolve my comments. I just have few comments for the authors' consideration as below:

1. The authors replied that an image analysis software which determined the snow level on the striped poles was used to determine snow height from photographs taken by a time-lapse camera. Please provide the software name and the associated citation.

*We now mention line 166 that "The image analysis software Fiji (https://imagej.net/software/fiji/ last accessed on 27 February 2025) was used to determine the snow level on the striped poles"*

2. Figure 3 caption: PWR; Power -> PWR: Power

*Thank you, changed.*

3. Figure 1 caption: Fig. 1 to Figure. 1.

*Thank you, changed*

4. The current figure quality need to be improved in terms of numbering, color and styles.

*As detailed in the response to the Editor's comments, Figures 6, 7, 8, 9 and 11 were modified.*

5. There are too many very small tables with just 1 line data, which can be improved

*We are surprised by this comment. Should a Table necessarily have many lines? Table 2 is a full page long, because it needs to. Table 5 just has one line of data, because that is all that needs to be shown. The small Tables are all different and cannot be regrouped.*

**Report 2 by Reviewer 1**

The authors have made only minor modifications to improve their manuscript. However, significant work remains before the manuscript can be considered suitable for acceptance.

As I and another reviewer have recommended, Section 3 is overly simplistic and requires more detailed information about the data and figures. The authors' response, stating that they "want to keep this simple and concise," is not a valid justification. If simplicity is the goal, they could simply list the figures and tables without any explanatory text within their manuscript. However, without sufficient description and interpretation of the data and results, the manuscript's quality does not meet the standards required for publication in The Cryosphere.

*Thank you, but this comment is not very helpful for lack of specificity. Addressing this comment would require constructive details or suggestions, without which no useful modification can be*

*made. The Editor has made some specific, clear and useful comments that we have done our best to address. We hope the resulting changes will satisfy the reviewer.*

Additionally, the term "snow depth" is the correct and professional terminology in this field. The authors MUST replace all instances of "snow height" with "snow depth". As defined by the National Snow and Ice Data Center (NSIDC), snow depth refers to "the combined total depth of both old and new snow on the ground" (source: NSIDC Cryosphere Glossary). This correction is essential for accuracy and consistency with established scientific terminology.

*Snow depth is a wonderful variable, when the snow surface is used as a reference. When we discuss radiation penetration, then of course the reference is the snow surface and we then use snow depth (e.g., lines 194, 212, 347, 389, 407, 416, 437). However, given that the sensors are at fixed height and certainly not at fixed depth, snow depth to discuss irradiance at the sensors' level is a totally useless variable in this case, and snow height obviously has to be used. We are shocked by the reviewer's insistence and especially by the inacceptable tone, which only shows the reviewer's ignorance of a large fraction of the snow literature, in particular the snow physics field, where snow height is often used for good reasons.*

*For the reviewer's information, if there is an internationally recognized snow terminology for snow physics work, it is "The International classification for seasonal snow on the ground" (Fierz et al., 2009), written by an international team of experts from many institutions, including Richard Armstrong from NSIDC, and not some glossary by one institution, which I have never seen, as there are anyway many such glossaries. (Fierz et al., 2009) mention the use of both snow height and snow depth, depending on the application. For snow pit work, snow height is often used, because the reference is the ground. Snow modelers also often use snow height to track a layer coordinate, rather than snow depth, for obvious reasons.*

*In summary, snow depth and snow height have to be used with judgement and not in a rigid and dogmatic manner. When we use snow height, it is because this is the sensible variable. We consider this discussion definitely closed.*

**Reference cited**

Fierz, C., Armstrong, R. L., Durand, Y., Etchevers, P., Greene, E., McClung, D. M., Nishimura, K., Satyawali, P. K., and Sokratov, S. A.: The International classification for seasonal snow on the ground UNESCO-IHP, ParisIACS Contribution N°1, 80 pp., 2009.

https://unesdoc.unesco.org/ark:/48223/pf0000186462